# Continuous Chain of Thought Enables Parallel Exploration and Reasoning

## Abstract

Current language models generate chain-of-thought traces by autoregressively sampling tokens from a finite vocabulary. While this discrete sampling has achieved remarkable success, conducting chain-of-thought with continuously-valued tokens (CoT2) offers a richer and more expressive alternative. Our work examines the benefits of CoT2 through logical reasoning tasks that inherently require search capabilities and provide optimization and exploration methods for CoT2. Theoretically, we show that CoT2 allows the model to track multiple traces in parallel and quantify its benefits for inference efficiency. Notably, one layer transformer equipped with CoT2 can provably solve the combinatorial "subset sum problem" given sufficient embedding dimension. These insights lead to a novel and effective supervision strategy where we match the softmax outputs to the empirical token distributions of a set of target traces. Complementing this, we introduce sampling strategies that unlock policy optimization and self-improvement for CoT2. Our first strategy samples and composes $K$ discrete tokens at each decoding step to control the level of parallelism, and reduces to standard CoT when $K = 1$. Our second strategy relies on continuous exploration over the probability simplex. Experiments confirm that policy optimization with CoT2 indeed improves the performance of the model beyond its initial discrete or continuous supervision.

## 1. Introduction

Chain-of-thought (CoT) strategies ([Wei et al., 2022](#)), when paired with strong base models, have achieved immense success and facilitated progress in remarkably challenging tasks, such as solving AIME or IOI problems ([Guo et al., 2025](#); [Jaech et al., 2024](#)). In essence, CoT boosts the expressive capability of the base model through autoregressive generation, a principle that also underlies the recent efforts on test-time compute scaling ([Snell et al., 2024](#)). Despite these advances, modern language model architectures may not fully utilize their potential for a few reasons. First is their discrete sampling of tokens—selecting a single token at each decoding step from a vocabulary of $v$ tokens. This limits the model to emitting at most $\log_2(v)$ bits per sample, or more specifically, the Shannon entropy of the softmax output. This contrasts with the $O(d)$ bits each token embedding can store, where $d$ is the embedding dimension. Secondly, discrete sampling can cause the model to *commit* to certain solutions and avoid exploring alternatives ([Yao et al., 2023](#)). A practical method to address this is sampling multiple CoT traces and aggregating them, either through consistency ([Wang et al., 2022](#)) or best-of-N decoding ([Ouyang et al., 2022](#)) through more test-time computation.

In this work, we propose and investigate the use of *CoT with Continuous Tokens* (CoT2) to address these challenges, building on COCONUT ([Hao et al., 2024](#)). The fundamental idea in our CoT2 proposal is that rather than the model sampling a single token from the vocabulary, it samples or deterministically selects a continuous superposition of tokens according to the softmax output. Intuitively, this capability—effectively selecting multiple tokens simultaneously through a continuous superposition—would allow the model to pack more information within each token embedding and also enable it to track multiple reasoning paths in parallel—potentially emulating self-consistency or best-of-N decoding with a single trace. Toward this vision, we make the following technical contributions:

- **Mechanistic and theoretical study of CoT2:** We quantify the benefits of CoT2 along two directions. First, we examine the problem of *Minimum Non-Negative Sum* (MNNS) as a generalization of the classical Subset Sum problem. These problems, as well as related tasks like ProntoQA ([Saparov & He, 2022](#)), inherently benefit from parallel search capability. We show that a single layer transformer can solve MNNS using CoT2, showcasing the capability of transformers to track and expand multi-

[1]Anonymous Institution, Anonymous City, Anonymous Region, Anonymous Country. Correspondence to: Anonymous Author <anon.email@domain.com>.

Preliminary work. Under review by the International Conference on Machine Learning (ICML). Do not distribute.

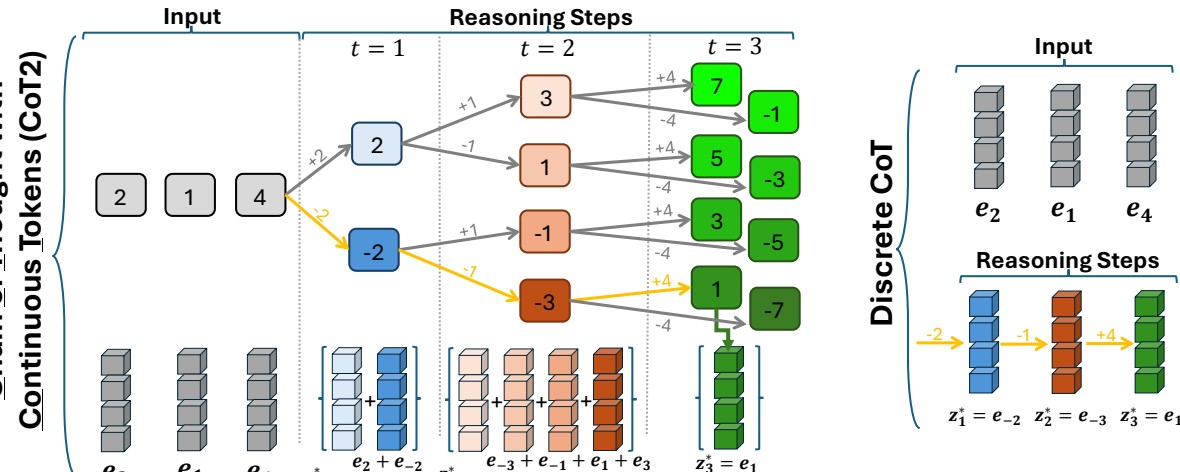

Figure 1: Illustration of CoT2 and discrete CoT for Minimum Non-Negative Sum (MNNS) task with $m = 3$. The input numbers are 2, 1, 4, and the correct path for this task $(-2, -3, 1)$ is highlighted with yellow arrows and corresponds to the discrete CoT supervision. CoT2 supervision for the reasoning steps $t \in \{1, \ldots, m - 1\}$ is the average of embeddings of reachable states, and for $t = m$ is the embedding corresponding to the answer.

ple reasoning traces in latent space. Complementing this, under a certain trajectory decoupling assumption, we provide a theoretical study of CoT2 decoding methods

- **Base CoT2:** deterministic inference which creates and feeds continuous tokens using full softmax output at each step (Sec. 2&3);
- **CoT2-MTS (multi-token sampling):** our method which samples $K$ discrete tokens from softmax and averages them to form a continuous token (Sec. 4);

and standard CoT which is a special case of CoT2-MTS with $K = 1$. We show that base CoT2 tracks and aggregates all reasoning paths whereas CoT2-MTS strictly generalizes CoT by tracking $K$ paths; and establish the sample complexity benefits of the CoT2 methods.

- **Supervision and reinforcement for CoT2:** We introduce the continuous supervision strategy CSFT for CoT2 models to explicitly track multiple teacher traces in parallel by fitting a target softmax map of the empirical distribution of tokens within the trace. Our method also reveals fundamental tradeoffs between the CoT2 accuracy and the embedding dimension. Complementing this, we introduce policy optimization methods for CoT2 (Section 4). We propose *MTS* as our primary strategy, which samples and composes K discrete tokens at each forward pass to control the level of parallelism. We also introduce a purely continuous sampling scheme over the probability simplex. Experiments on the MNNS, ProntoQA, and ProsQA tasks demonstrate that GRPO-based RL with CoT2 further improves the accuracy over SFT or CSFT (see Section 4.3). This demonstrates that the RL phase helps the model better prioritize relevant reasoning traces

and unlocks a promising strategy for training CoT2-based language models.

Ultimately, our results and methods underscore the strong potential of CoT2 and encourage further research. The rest of the paper is organized as follows: Section 2 introduces the technical setup, Section 3 describes our continuous supervision strategy as well as the MNNS, ProntoQA, and ProsQA tasks. Section 4 describes our sampling strategies and the resulting GRPO-based policy optimization methods. Section 5 provides theoretical guarantees and Section 6 concludes with a discussion.

### 1.1. Related Work

The efficacy of eliciting reasoning in LLMs through chain-of-thought (CoT) prompting has been well-established (Nye et al., 2021; Wei et al., 2022; Kojima et al., 2022; Suzgun et al., 2023; Guo et al., 2025). CoT prompting provides a convenient way to increase inference-time compute and computational depth, both of which have been found to be independently useful (Pfau et al., 2024; Goyal et al., 2024; Feng et al., 2023; Merrill & Sabharwal, 2024). However, the discrete nature of CoT tokens forces sequential exploration of reasoning paths, resulting in longer reasoning paths and consequently increased inference-time compute. Furthermore, restricting reasoning to natural language can be inefficient, as groups of tokens can often be more effectively represented by a single continuous token. Thus, CoT2 offers an alternative strategy for compute-efficient reasoning and complements methods that aim to shorten/control the trace length of CoT (Aggarwal & Welleck, 2025; Zhang et al., 2025; Sui et al., 2025).

One way to address these challenges is by leveraging the implicit reasoning capabilities of transformers (Yang et al., 2024; Shalev et al., 2024). Works such as (Deng et al., 2023; 2024; Yu et al., 2024) use various techniques to obtain models that can perform reasoning internally without emitting CoT tokens. Another line of work has found looped transformers to be effective on reasoning problems (Giannou et al., 2023; Geiping et al., 2025), notably being able to mimic CoT (Saunshi et al., 2025) with a sufficient number of iterations. Our work is similar to this line of work in that continuous representations are used to perform reasoning.

Our work is most related to a recent body of work introducing LLMs capable of reasoning with explicit continuous tokens decoded autoregressively. In particular, recently proposed COCONUT (Hao et al., 2024) autoregressively feeds the last token's final-layer representation as input to the next step. Given labeled CoT data, COCONUT is trained to progressively replace discrete tokens with continuous tokens (from left to right). Shen et al. (2025) propose CODI, where an LLM with continuous CoT is supervised to produce the correct answer, while also aligning its hidden representation on the last reasoning token to that of a discrete CoT model that shares the same backbone. Cheng & Van Durme (2024) propose CCOT, where an auxiliary module is first trained to decode autoregressively a compressed representation of a discrete CoT trace, and later the main LLM is fine-tuned to produce correct answers by additionally conditioning on the generated continuous tokens. While COCONUT, CODI, CCOT, and our CoT2 all aim to reason in continuous space, we propose distinct algorithmic approaches that also address the exploration challenge. Key differences include: (1) Our continuous tokens are simplex-weighted compositions of vocabulary tokens. (2) Our supervision method is novel and explicitly targets implicit parallelism. (3) CoT2 does not initialize from, nor attempt to mimic, discrete CoT. (4) By introducing sampling strategies and associated GRPO variations, we realize the "Supervised Training → Reinforcement Learning" paradigm in the context of CoT2. We provide further discussion of literature on multi-token prediction and reinforcement learning in Appendix A.

## 2. Problem Setup

**Notation.** For an integer $n \geq 1$, we use the shorthand $[n] = \{1, \ldots, n\}$. Throughout, we denote vectors by bold lowercase letters (e.g. $\boldsymbol{x}$) and matrices by bold uppercase letters (e.g. $\boldsymbol{X}$). For a vector $\boldsymbol{x} \in \mathbb{R}^n$, the component $x_i$ refers to its $i$-th entry. The zero vector in $\mathbb{R}^n$ is written as $\boldsymbol{0}_n$, and the zero matrix in $\mathbb{R}^{m \times n}$ is $\boldsymbol{0}_{m \times n}$. Finally, we let $\Delta^{v-1}$ denote the standard $v - 1$ simplex in $\mathbb{R}^v$.

Assume that we are given an input context $\boldsymbol{X} \in \mathbb{R}^{n \times d}$, where each of the $n$ rows is a $d$-dimensional embedding vector. Our objective is to output $m$ tokens given the context $\boldsymbol{X}$ with

$m$th output token being the final answer that is evaluated under some performance metric (e.g. accuracy or reward). For the first $m-1$ steps, the model outputs *continuous* tokens $\{z_t\}_{t \in [m-1]}$, which are *thought tokens* that enable a reasoning process. At the final step $t = m$, the model outputs a *discrete* token $z_m$ from a vocabulary of size $v$. In the remainder of this paper, we investigate strategies for training this system in a way that improves final performance over standard discrete next-token prediction.

Formally, let $\boldsymbol{E} = [\boldsymbol{e}_1, \ldots, \boldsymbol{e}_v]^\top \in \mathbb{R}^{v \times d}$ be the embedding matrix corresponding to the vocabulary of $v$ tokens, where $\boldsymbol{e}_i \in \mathbb{R}^d$ represents the embedding of the $i$th token. We define the next-token prediction model $\text{LM}_\theta$ parameterized by $\theta$ that assigns, at each step $t$, a probability distribution over possible next tokens given the prefix $z_{<t}$ and context $\boldsymbol{X}$. Concretely, for $1 \leq t \leq m - 1$, the model outputs the following probability distribution over the $v$ vocabulary entries via a softmax operation:

$$\text{LM}_\theta(\cdot \mid z_{<t}, \boldsymbol{X}) := \boldsymbol{\alpha}_t \text{ where}$$

$$\boldsymbol{\alpha}_t = \left[ \alpha_{t,1}, \ldots, \alpha_{t,v} \right] \in \Delta^{v-1}, \text{ i.e. } \alpha_{t,i} \geq 0 \text{ and } \sum_{i=1}^{v} \alpha_{t,i} = 1$$

We then form the continuous token as the convex combination of all tokens in the vocabulary:

$$z_t = \boldsymbol{E}^\top \boldsymbol{\alpha}_t \in \mathbb{R}^d, \quad \forall 1 \leq t \leq m - 1$$

Hence each continuous token $z_t$ is a linear combination of the vocabulary embeddings. At the final step $t = m$, the model samples a discrete token $z_m \in \{\boldsymbol{e}_1, \ldots, \boldsymbol{e}_v\}$ from its policy distribution $\text{LM}_\theta(\cdot \mid z_{<m}, \boldsymbol{X}) = \boldsymbol{\alpha}_m$. Finally, we note that we assume that the answer depends only on the final discrete token $z_m$ merely for simplicity; the same framework naturally extends to decoding multiple final discrete tokens after continuous ones. We refer to this decoding strategy as **base CoT2** and observe that it results in a deterministic reasoning chain because the continuous tokens are precisely determined by the softmax map. In Section 4, we will introduce stochastic alternatives, such as **CoT2-MTS**, to facilitate generative reasoning.

## 3. CSFT: A Supervised Training Method for CoT2

In this section, we present our method of continuous supervised training to learn intermediate thought tokens as "soft" targets rather than "hard" target tokens, as described in Section 2. Specifically, we provide the model with convex combinations of vocabulary embeddings, which allows the model flexibility in those reasoning steps. Such an approach is particularly suitable when the task accuracy depends only on the final token or token distribution. Formally, at each reasoning step $t = 1, \ldots, m - 1$, the supervision specifies a

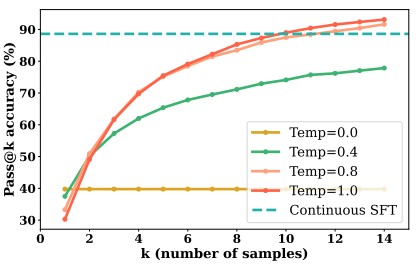 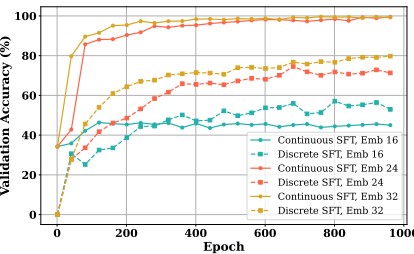 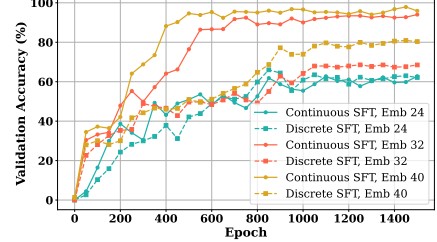

(a) Continuous vs. discrete SFT (Pass@k) accuracies for different temperatures on MNNS task.

(b) Training performance vs. embedding dimension for continuous and discrete SFT on MNNS task.

(c) Training performance vs. embedding dimension for continuous and discrete models on ProsQA.

Figure 2: The teacher distribution for the continuous model is derived from a search algorithm. **(a):** The figure illustrates that the discrete model requires multiple samplings (Pass@k) to match the single-sample performance of the continuous model on MNNS (10-run average). **Setting:** 4 input digits in $1 - 9$; 1-layer, 1-head GPT2 with $d = 24$. **(b-c):** The figures reveal that above a certain embedding dimension threshold, the continuous model is superior to discrete in tasks involving search, like MNNS and ProsQA. **Setting (b):** 4 input digits in $1 - 9$; 2-layer, 2-head GPT2 with $d \in \{16, 24, 32\}$. **(c):** 4-layer, 4-head GPT2 with $d \in \{24, 32, 40\}$.

target probability distribution

$$\boldsymbol{\alpha}_t^* = \left[ \alpha_{t,1}^*, \ldots, \alpha_{t,v}^* \right] \in \Delta^{v-1},$$

where $\alpha_{t,i}^* \geq 0$ and $\sum_{i=1}^{v} \alpha_{t,i}^* = 1$. We train the model to align its predicted distribution $\boldsymbol{\alpha}_t$ to the supervision distribution $\boldsymbol{\alpha}_t^*$ rather than one-hot labels, with the help of a divergence-based loss:

$$\mathcal{L}_{\text{cont}}(\theta; \boldsymbol{X}, t) = D\left( \boldsymbol{\alpha}_t^* \,\|\, \boldsymbol{\alpha}_t \right),$$

where $D(\cdot\|\cdot)$ is the cross-entropy (or equivalently KL divergence) between two distributions. This approach can also be viewed as *token-level knowledge-distillation*, where the teacher distribution $\boldsymbol{\alpha}_t^*$ may be obtained through a logic/search algorithm. At the final step $t = m$, we typically have a discrete target $z_m^* \in \{e_1, \ldots, e_v\}$, so that $\boldsymbol{\alpha}_m^*$ is one-hot distribution placing probability 1 on that target token and 0 elsewhere. This is equivalent to employing a standard cross-entropy loss $-\log \text{LM}_\theta\left(z_m^* \mid \boldsymbol{z}_{<m}, \boldsymbol{X}\right)$ at the final step. Hence, for each training example, the total loss for the proposed continuous supervised training is the sum of the continuous-token divergence losses:

$$\mathcal{L}_{\text{CSFT}}(\theta; \boldsymbol{X}) = \sum_{t=1}^{m} \mathcal{L}_{\text{cont}}(\theta; \boldsymbol{X}, t) \qquad (1)$$

By minimizing $\mathcal{L}_{\text{CSFT}}(\theta)$, we teach the model to learn the soft targets $\boldsymbol{\alpha}_t^*$ at each step and to predict the correct final discrete token. In the above training procedure, inspired by the discussions in Bachmann & Nagarajan (2024); Bengio et al. (2015), we consider two ways of providing prefixes to the language model:

1. **Teacher forcing:** Each step $t$ is conditioned on the ground-truth prefix $\boldsymbol{z}_{<t}^*$, meaning that the model has access to all ground-truth previous tokens during prediction. Concretely, for each step $t' < t$, the corresponding input $\boldsymbol{z}_{t'}^* = \boldsymbol{E}^\top \boldsymbol{\alpha}_{t'}^*$ is a convex combination of all vocabulary tokens.

2. **Self-feeding:** Each step $t$ autoregressively uses the model's previously generated outputs, $\boldsymbol{z}_{<t}$, during training. In particular, as described in Section 2, the continuous output token $\boldsymbol{z}_t = \boldsymbol{E}^\top \boldsymbol{\alpha}_t$, is a convex combination of vocabulary embeddings, which is then fed back to the model as part of the prefix.

It is also worth noting that one may apply *temperature scaling* or *thresholding* to $\boldsymbol{\alpha}_t$ before forming $\boldsymbol{z}_t$ in order to filter the model's predictions. In our experiments, we find that teacher forcing leads to superior performance for CSFT, even though at inference time, the model runs in an autoregressive manner, as discussed below. See Appendix C for further discussion.

**Inference.** At inference time, the model does not rely on the ground-truth distributions $\boldsymbol{\alpha}_t^*$. Instead, at each continuous step $t < m$, the model autoregressively uses its own output distribution $\boldsymbol{\alpha}_t$ by converting that distribution to a continuous token $\boldsymbol{z}_t = \boldsymbol{E}^\top \boldsymbol{\alpha}_t$ and adds it to the prefix for the prediction in the next step. At the final step, the model generates a discrete sample from $\boldsymbol{\alpha}_m = \text{LM}_\theta(\cdot \mid \boldsymbol{z}_{<m}, \boldsymbol{X})$.

**Discrete baseline.** In this case, we use teacher-forced training where the next token prediction is performed conditioned on the previous ground-truth tokens with standard cross-entropy loss. Discrete baseline enforces $\boldsymbol{z}_t^*$ to be a token in vocabulary $\{e_1, \ldots, e_v\}$, which means that it is a special case of CSFT where the $\boldsymbol{\alpha}_t^*$ are one-hot vectors rather than an arbitrary element of $\Delta^{v-1}$. The model minimizes the following objective, which is obtained by summing over all steps of teacher-forced next-token prediction:

$$\mathcal{L}_{\text{SFT}}(\theta; \boldsymbol{X}) = \sum_{t=1}^{m} -\log \text{LM}_\theta\left(z_t^* \mid z_{<t}^*, \boldsymbol{X}\right). \qquad (2)$$

## 3.1. Tasks Requiring Exploration over States

In the next two subsections, we illustrate the CSFT training described in (1) on tasks that require *exploration* over multiple states. Consider that the vocabulary is sufficiently large that each state $g$ of the task can be assigned a unique embedding. Then, we let $\Gamma_t$ be the set of all states that could result from building upon step $(t-1)$, where $\Gamma_0 = \{g_0\}$ for the initial state $g_0$. For each element $g \in \Gamma_t$, we assign a probability $\alpha_{t,g}^*$ that reflects how many times that state occurs under a search process. Then, $\alpha_t^*$ is formed by normalizing these probabilities into a distribution on $\Gamma_t$:

$$\alpha_{t,g}^* = \frac{\text{count}_t(g)}{\sum_{h \in \Gamma_t} \text{count}_t(h)}, \qquad (3)$$

where $\text{count}_t(g)$ is the number of times state $g$ appears among all expansions at step $t$. At the final step $t = m$, we select exactly one correct state from $\Gamma_m$, so that $\alpha_m^*$ is a one-hot vector:

$$\alpha_{m,g}^* = \begin{cases} 1, & \text{if } g \text{ is the correct final state,} \\ 0, & \text{otherwise.} \end{cases}$$

**Remark.** While we focus on search-based tasks MNNS and ProntoQA in the next two sections, one can also extend to training with continuous tokens in the language-model context. The distributions $\{\alpha_t^*\}$ at each step can be collected by (1) running a beam or best-first search to generate multiple partial trajectories; (2) scoring these trajectories with a reward function; and (3) curating them into a distribution that assigns higher mass to states that leads to higher rewards. This construction of $\alpha_t^*$ replaces one-hot supervision with soft supervision for intermediate reasoning steps.

### 3.1.1. Minimum Non-Negative Sum Task

We now introduce the *Minimum Non-Negative Sum* (MNNS) task, where the goal is to assign signs to a list of numbers so that their sum is as small as possible while being nonnegative. The MNNS task can be viewed as partitioning a set of numbers into two subsets with a minimal difference, which makes it closely related to the subset sum problems explored in Dziri et al. (2023); Thomm et al. (2024). Formally, we are given $m$ integers $d_1, \ldots, d_m$, and the task is to assign signs $\sigma_i \in \{+1, -1\}$ such that $s = \sigma_1 d_1 + \cdots + \sigma_m d_m \geq 0$ and $s$ is minimized. Let $\sigma^{\text{opt}} = (\sigma_1^{\text{opt}}, \ldots, \sigma_m^{\text{opt}})$ denote the optimal assignment that achieves the minimal nonnegative sum $s^{\text{opt}}$ out of $2^m$ possible sign assignments. Here, every possible *partial sum* $\sigma_1 d_1 + \cdots + \sigma_t d_t \in \Gamma_t$ is assigned a unique embedding $e_{\phi(\sigma_1 d_1 + \cdots + \sigma_t d_t)}$, where $\phi(\cdot)$ maps each sum to a distinct id in $[v]$. We now describe the two modes of supervision, where the input digits are processed one by one by accumulating partial sums, as illustrated in Figure 1:

- *Supervision for CoT2 model:* At step $t$, there are $|\Gamma_t| = 2^t$

partial sums of length $t$, and accordingly, we provide the following target distribution $\alpha_t^*$:

$$\alpha_{t,i}^* = \begin{cases} \dfrac{\text{count}_t(i)}{2^t}, & \text{if token } i \text{ appears } \text{count}_t(i) \text{ times} \\ & \quad \text{as a partial sum of length } t; \\ 0, & \text{otherwise.} \end{cases}$$

At the final step $t = m$, the distribution $\alpha_m^*$ assigns probability 1 to the correct sum $e_{\phi(\sigma_1^{\text{opt}} d_1 + \cdots + \sigma_m^{\text{opt}} d_m)}$ and 0 to all others.

- *Supervision for discrete model:* We supervise the discrete model along the correct chain of partial sums by providing $e_{\phi(\sigma_1^{\text{opt}} d_1 + \cdots + \sigma_t^{\text{opt}} d_t)}$ for $1 \leq t \leq m$ as target tokens, and train following the standard cross-entropy objective described in (2).

While constructing the dataset, we split the training and validation sets by ensuring that any permutation of numbers appears in exactly one split. The aim behind this is to prevent memorization and make a fair evaluation. We also encode input and output numbers with separate tokens in our vocabulary. As an example, an input appears as $\langle \text{BOS} \rangle$ $d_1$ $d_2$ $\ldots$ $\rightarrow$, and the corresponding output as $s_1$ $s_2$ $\ldots$ $s^{\text{opt}}$ $\langle \text{EOS} \rangle$, where $s^{\text{opt}}$ is the minimal nonnegative sum for $\{d_1, \ldots, d_m\}$. For the model, we use the GPT2 architecture (Radford et al., 2019) with different head, layer, and embedding dimension configurations, and train it from scratch. Please refer to Appendix B for more experimental details.

### 3.1.2. ProntoQA and ProsQA Datasets

Other datasets we explore in our investigation of the CSFT approach are the ProntoQA (Saparov & He, 2022) and ProsQA (Hao et al., 2024) datasets, which are logical reasoning tasks that require exploration over multiple possible paths. Specifically, each question in ProntoQA asks whether a certain target word (node) $B$ is reachable from a root word (node) $A$ within a fixed number of hops, while for ProsQA it asks which of the target words $B$ or $C$ is reachable. We use 5-hop questions and present the graph in a structured format. In particular, for each problem, we represent nodes and edges using embeddings, which we use as the model input rather than text input. The detailed structured format and examples are provided in Appendix B.2.

The graph structure of the ProntoQA and ProsQA datasets naturally obeys the supervision in (3), so that we determine the words that can be reached using $t$ edges from $A$ and supervise intermediate tokens on the resulting distribution. At the final reasoning step $m$, the supervision assigns probability 1 to the correct label: `yes` or `no` for ProntoQA, and `B` or `C` for ProsQA. For the standard discrete model, we provide an explicit chain of nodes from $A$ to the target node

($B$ or $C$) as the target path at each step. We direct readers to Appendix B.2 for additional details on the supervision.

## 3.2. Results and Discussion of CoT2 Supervision

In our experiments on the MNNS, ProsQA, and ProntoQA tasks, we observe that CSFT significantly outperforms discrete baseline once the embedding dimension exceeds a moderate threshold, as shown in Figures 2b and 2c. We also note that continuous tokens allow for a faster convergence above this threshold, as illustrated by Figures 2b and 2c. Through continuous tokens, the model gains the capacity to represent multiple partial expansions in parallel, enabling a 'search-like' ability that results in higher accuracy. We also demonstrate in Figure 2a that the discrete model requires multiple sampling (pass@k) to approach the performance achieved by a single attempt of the CoT2 model. This indicates that continuous tokens effectively hedge against early mistakes, as there's no accumulating error. This argument aligns with the previous "snowballing errors" discussions for discrete autoregressive generation in (Bachmann & Nagarajan, 2024). Moreover, although the continuous approach needs more embedding capacity to allow its distributional representations at each step, it can then achieve strong performance with fewer layers and heads compared to the discrete model, as further demonstrated by the results in Appendix C.1. We also provide experiments on ProntoQA in Appendix C.1, which confirm similar findings to results on ProsQA.

We also evaluated the performance of the discrete model under more sparse supervision scenarios, such as providing only a subset of the correct partial sums or, more extremely, only the final answer. In these experiments, we observed that denser supervision improved the discrete CoT model's performance; see Appendix C.1 for further details.

## 4. Reinforcement Learning Methods for CoT2

In this section, we describe how to apply RL with continuous output tokens. Specifically, we explore GRPO training on top of continuous or discrete models that are supervised trained based on Section 3 for the MNNS, ProntoQA, and ProsQA tasks. By illustrating two sampling methods for GRPO, we demonstrate how a model trained with discrete SFT can be adapted to produce continuous outputs. We assume a sparse reward setting where the reward is 1 for a correct final answer and 0 otherwise.

In our setup, a language model $LM_\theta$ acts as a *policy* over tokens. Let $\{\mathbf{Z}^{(i)}\}_{i=1}^{G}$ be a group of $G$ trajectories sampled from old policy $LM_{\theta_{old}}$ such that each trajectory $\mathbf{Z}^{(i)} = \left(z_1^{(i)}, \ldots, z_m^{(i)}\right)$ contains $m$ output tokens given a fixed input $\mathbf{X}$. We denote by $\hat{A}_{i,t}$ the advantage estimate at step $t$ in trajectory $i$ and note that $\hat{A}_{i,t} = \hat{A}_i$ is identical across

---

**Algorithm 1** Multi-Token Sampling GRPO for Continuous Token Generation

**Input:** Initial policy $LM_{\theta_{init}}$; hyperparameters $K, G, m, \epsilon, \beta$.
1: $LM_\theta, LM_{\theta_{ref}} \leftarrow LM_{\theta_{init}}$
2: **for** iteration = 1, 2, ..., I and **for** step = 1, 2, ..., S **do**
3:     Sample a batch of inputs $\{\mathbf{X}^{(b)}\}_{b=1}^{B}$
4:     Update $LM_{\theta_{old}} \leftarrow LM_\theta$
5:     **for** each input $\mathbf{X}$ in the batch and **for** each trajectory $i = 1, \ldots, G$ from that $\mathbf{X}$ **do**
6:         **for** each token step $t = 1, \ldots, m$ **do**
7:             **if** $t < m$ **then**
8:                 Sample $K$ tokens $\{\mathbf{e}_{i_1}, \ldots, \mathbf{e}_{i_K}\}$ from $\alpha_t^{(i),old}$ to create continuous token $z_t \leftarrow \frac{1}{K} \sum_{r=1}^{K} \mathbf{e}_{i_r}$.
9:                 Policy ratio $r_t(\theta) \leftarrow \left(\prod_{r=1}^{K} \alpha_{t,i_r}^{(i)} / \prod_{r=1}^{K} \alpha_{t,i_r}^{(i),old}\right)^{\frac{1}{K}}$.
10:             **else**
11:                 Sample $z_m = \mathbf{e}_j$ from $\alpha_m^{(i),old}$.
12:                 Policy ratio for discrete token $r_m(\theta) \leftarrow \alpha_{m,j}^{(i)} / \alpha_{m,j}^{(i),old}$.
13:             **end if**
14:         **end for**
15:         Obtain advantage estimates $\hat{A}_{i,t}$ for each token $t$ in each trajectory $\mathbf{Z}^{(i)}$ and calculate objective.
16:     **end for**
17:     Update $\theta$ to minimize $\mathcal{L}_{GRPO}(\theta)$.
18: **end for**
**Output:** $LM_\theta$

---

all steps of a trajectory under sparse reward setting. To quantify how the new policy $LM_\theta$ differs from the old one on token $z_t^{(i)}$ from $i$th trajectory, we define the policy ratio $r_t^{(i)}(\theta) = \frac{LM_\theta\left(z_t^{(i)} | z_{<t}^{(i)}, \mathbf{X}\right)}{LM_{\theta_{old}}\left(z_t^{(i)} | z_{<t}^{(i)}, \mathbf{X}\right)}$. We update the model parameters $\theta$ by minimizing the clipped surrogate objective (Shao et al., 2024; Yu et al., 2025):

$$\mathcal{L}_{GRPO}(\theta) = -\frac{1}{\sum_{i=1}^{G} |\mathbf{Z}^{(i)}|} \sum_{i=1}^{G} \sum_{t=1}^{|\mathbf{Z}^{(i)}|} \Big[\min\left(r_t^{(i)}(\theta) \hat{A}_{i,t},\right.$$
$$\left. clip\left(r_t^{(i)}(\theta), 1-\epsilon, 1+\epsilon\right) \hat{A}_{i,t}\right) - \beta \mathbb{D}_{KL}\left[LM_\theta \| LM_{\theta_{ref}}\right]\Big].$$

As the output length is fixed in our setting, we have $|\mathbf{Z}^{(i)}| = m$ for each trajectory. Here, $\epsilon$ is a clipping parameter that bounds the ratio $r_t(\theta)$, and $\beta$ controls the strength of KL-divergence from a reference policy $LM_{\theta_{ref}}$ which is the SFT-initialized policy. We set the number of GRPO iterations $\mu = 1$ and estimate the KL divergence with the Schulman Approximator as in Shao et al. (2024).

### 4.1. Multi-Token Sampling

We emulate the rollout of a continuous token by sampling a fixed number of $K \leq v$ discrete tokens and averaging them at steps $t = 1, \ldots, m-1$. We refer to this hybrid method as *CoT2-MTS* (multi-token sampling). For the GRPO objective, we propose the following method to calculate the policy ratio for continuous tokens. Specifically, assume at step $t$ we sample discrete tokens $\mathbf{e}_{i_1}, \ldots, \mathbf{e}_{i_K}$ with probabilities $\alpha_{t,i_1}, \ldots, \alpha_{t,i_K}$ under the current policy and probabilities $\alpha_{t,i_1}^{old}, \ldots, \alpha_{t,i_K}^{old}$ under the old policy. We define the policy ra-

Table 1: Validation accuracy and token-level entropy of CoT2-MTS sampling GRPO on the discrete model under different rollout sizes $K$ for MNNS task. We use 4 input digits in 1-9 with 1-layer, 1-head GPT2 at embedding dimensions 24 and 32, with SFT accuracies of 39.76% and 43.50%, respectively.

| | $K$ | Val. Accuracy (%) | | Val. Entropy (SFT → SFT+GRPO) | | | |
| | | SFT | SFT+GRPO | token$_1$ | token$_2$ | token$_3$ | token$_4$ |
|---|---|---|---|---|---|---|---|
| 24 | 1 | | 49.01 | 0.3218 → 0.0314 | 0.5858 → 0.0712 | 0.5499 → 0.1576 | 0.4786 → 0.1718 |
| | 3 | 39.76 | 52.60 | 0.3717 → 0.0647 | 0.7461 → 0.2120 | 0.8006 → 0.3312 | 0.5338 → 0.1529 |
| | 6 | | 49.69 | 0.4524 → 0.1242 | 0.7738 → 0.3361 | 0.8364 → 0.6615 | 0.5134 → 0.2159 |
| 32 | 1 | | 51.61 | 0.3618 → 0.0143 | 0.6331 → 0.0395 | 0.3518 → 0.2631 | 0.1962 → 0.1182 |
| | 3 | 43.50 | 55.66 | 0.3886 → 0.0412 | 0.7101 → 0.0904 | 0.5447 → 0.5757 | 0.2863 → 0.1734 |
| | 6 | | 50.38 | 0.4224 → 0.0632 | 0.7915 → 0.2161 | 0.6077 → 0.8481 | 0.2780 → 0.1514 |

tio for these continuous steps by dividing geometric means:

$$r_t(\theta) = \frac{\text{LM}_\theta\left(z_t \mid z_{<t}, X\right)}{\text{LM}_{\theta_{\text{old}}}\left(z_t \mid z_{<t}, X\right)} = \left(\frac{\alpha_{t,i_1} \cdots \alpha_{t,i_K}}{\alpha^{\text{old}}_{t,i_1} \cdots \alpha^{\text{old}}_{t,i_K}}\right)^{1/K}, \quad (4)$$

for $t = 1, \ldots, m - 1$. The geometric mean ensures that the ratio for each continuous step remains on the same scale as the final discrete token's ratio and, thus, helps avoid overly large or small updates in the GRPO objective and provides more stable training compared to the direct multiplication of probabilities. Once this ratio is computed, we then average the $K$ sampled tokens to form $z_t$, which is fed to the model as the query for the next prediction step. At the final step $t = m$, where the token $z_m = e_j$ is discrete with $j \in [v]$ denoting its index, the policy ratio is simply the probability ratio of selecting that token:

$$r_m(\theta) = \frac{\text{LM}_\theta\left(z_m \mid z_{<m}, X\right)}{\text{LM}_{\theta_{\text{old}}}\left(z_m \mid z_{<m}, X\right)} = \frac{\alpha_{m,j}}{\alpha^{\text{old}}_{m,j}}. \quad (5)$$

**Inference.** During inference after GRPO training, we apply the same multi-token sampling procedure at each of the first $m - 1$ steps to form the continuous token via the average of $K$ sampled embeddings.

**Remark.** An alternative to the normalization of ratios given by (4) is to directly scale down the logits by $1/K$ before applying softmax. However, using this approach during inference leads to a distribution shift relative to the SFT-trained model and ultimately degrades performance.

### 4.2. Dirichlet Sampling

In this section, we present another method for generating continuous tokens at each step by interpreting the model's output distribution $\alpha_t \in \Delta^{v-1}$ as concentration parameters of a Dirichlet distribution over the $v - 1$ simplex. We introduce a scaling hyperparameter $\gamma > 0$ and define the Dirichlet distribution with the parameters $\gamma \alpha_t = (\gamma \alpha_{t,1}, \ldots, \gamma \alpha_{t,v})$. Without this scaling, directly using $\alpha_t$ as parameters often

causes training instability, particularly when many $\alpha_{t,i}$ values are small. We then sample a point $\hat{\alpha}_t \in \Delta^{v-1}$ from the resulting distribution $\text{Dir}(\alpha_t)$. After sampling, we form the continuous token by mapping $z_t = E^\top \hat{\alpha}_t \in \mathbb{R}^d$, which becomes the query for the next step. We denote the Dirichlet densities induced by current and old policies as $f_\theta(z_t)$ and $f_{\theta_{\text{old}}}(z_t)$, respectively. Accordingly, we define the policy ratio at a continuous step $t < m$ as:

$$r_t(\theta) = \frac{\text{LM}_\theta(z_t \mid z_{<t}, X)}{\text{LM}_{\theta_{\text{old}}}(z_t \mid z_{<t}, X)} = \frac{f_\theta(z_t)}{f_{\theta_{\text{old}}}(z_t)},$$

The above definition parallels how we compute probability ratios for discrete actions but replace the categorical pmf with continuous Dirichlet pdf. At the final step $t = m$, we sample a discrete token $z_m \in \{e_1, \ldots, e_v\}$ from $\alpha_m$, and use the standard policy ratio given by (5). At inference, we follow the autoregressive procedure in Section 3 by creating a convex combination of vocabulary tokens.

### 4.3. Results and Discussion of Policy Optimization for CoT2

**MNNS evaluation:** Table 1 provides our results for the MNNS task and demonstrates that, for each $K \in \{1, 3, 6\}$, CoT2-MTS significantly improves validation accuracy relative to the discrete SFT baseline (39.76%), with moderate $K$ yielding the best final performance. We also observe that smaller $K$-values correspond to larger reductions in token-level entropies, suggesting that the model becomes more confident at each intermediate step by learning to commit to fewer tokens. This suggests a curriculum on $K$—starting small and gradually increasing—could potentially further improve the training on the MNNS task. Interestingly, the third token's entropy remains relatively high, which might indicate that the model continues to hedge among several partial expansions at that step, which may help preserve useful diversity of reasoning. Therefore, CoT2-MTS enables a model trained with discrete SFT to produce continuous outputs and helps it achieve better performance. Finally, Ap-

Table 2: Validation accuracies on ProsQA and ProntoQA for CoT2 and Discrete CoT, evaluated at $K = 6, 8, 12$ using CoT2-MTS sampling GRPO. All models use a 4-layer, 4-head GPT2 with embedding dimension 32. The SFT values are constant as they represent initial accuracies before GRPO. Remarkably, GRPO with our multi-token sampling scheme results in consistent improvements.

|  |  | ProsQA | | ProntoQA | |
|---|---|---|---|---|---|
|  |  | SFT | SFT+GRPO | SFT | SFT+GRPO |
| K=6 | CoT2 | 93.37 | 93.83 | 75.36 | 76.15 |
|  | Discrete CoT | 68.50 | 68.24 | 59.58 | 62.28 |
| K=8 | CoT2 | 93.37 | 94.09 | 75.36 | 76.66 |
|  | Discrete CoT | 68.50 | 71.58 | 59.58 | 71.53 |
| K=12 | CoT2 | 93.37 | 94.21 | 75.36 | 77.64 |
|  | Discrete CoT | 68.50 | 72.76 | 59.58 | 74.03 |

pendix C.2 shows that the CoT2 model with CSFT achieves a strong performance once the embedding dimension is sufficiently large (compared to the results in Table 1), however, it can be further improved with GRPO with Dirichlet Sampling.

**ProsQA and ProntoQA evaluation:** Table 2 provides an evaluation of the benefits of GRPO with CoT2-MTS on models trained with discrete or continuous SFT. **Remarkably, we observe that both CoT2 and discrete CoT models consistently improve across all rollout sizes** ($K = 6, 8, 12$), with larger $K$ values yielding better final accuracies by promoting more exploration. Notably, the discrete CoT model benefits relatively more from reinforcement learning compared to CoT2, likely because the CoT2 model already incorporates exploration implicitly through continuous supervision (CSFT). Aligning with this, we observe that for the ProntoQA dataset, the final performance of discrete CoT approaches that of the CoT2 model. Finally, we expect that, the performance gain of the MNNS task could be more limited compared to ProsQA and ProntoQA due to the highly structured nature of the MNNS task which makes CSFT supervision a natural choice and difficult to improve over.

## 5. Theoretical Analysis

In this section, we first present the construction of a single-layer transformer that solves the MNNS task using an attention layer followed by a mixture-of-experts MLP layer. We then provide a theoretical comparison between base CoT2, CoT2-MTS, and discrete CoT models.

### 5.1. Solving the Minimum Non-Negative Sum Task

**Proposition 1** (Solving MNNS)**.** *There exists a 1-layer transformer architecture with a mixture-of-experts MLP layer that solves the MNNS task using CoT2 by storing (sine,*

*cosine) embeddings of all $2^k$ states at the k-th iteration in a non-overlapping manner.*

The above construction utilizes trigonometric embeddings, inspired by the mechanistic insights given by Nanda et al. (2023). Our approach leverages these trigonometric embeddings to provide a theoretical guarantee that **the transformer can track and add/subtract multiple numbers in parallel** by benefiting from the embedding capacity and reading off the minimum non-negative number at the final step. An important observation regarding our construction is that the trajectories at each intermediate reasoning step are truly decoupled as it stores each state using non-overlapping (sine, cosine) representations. This is similar to the left side of Figure 1, but we also utilize rotations/shifts to ensure distinct states are orthogonal and are easy to read out.

### 5.2. Understanding and Formalizing the Benefits of CoT2 and Comparison to CoT

To proceed, we argue that CoT2 equips the model with the ability to track multiple paths in parallel which can be formalized through Assumption 1 below. Building on this condition, we will provide a formal comparison of CoT2 and discrete CoT models in the remainder of this section.

**Assumption 1.** *Recall the model $\mathrm{LM}_\theta$ in Section 2. For any step t and prefix tokens $z_{\leq t}$, we assume (i) the next token probabilities depend only on the last token $z_t$ and the query $X$ and (ii) if the last token is $z_t = \sum_{j=1}^v \alpha_{t,j} e_j$ so that $\sum_{j=1}^v \alpha_{t,j} = 1$, the output distribution $\alpha_{t+1}$ decouples as follows:*

$$\mathrm{LM}_\theta^{(t)}(\cdot \mid z_t, X) \overset{(i)}{=} \mathrm{LM}_\theta^{(t)}(\cdot \mid z_{\leq t}, X) \overset{(ii)}{=} \sum_{j=1}^v \alpha_{t,j} \mathrm{LM}_\theta^{(t)}(\cdot \mid e_j, X).$$

Under Assumption 1, the token distribution $\alpha_{t+1} = \mathrm{LM}_\theta^{(t)}(\cdot \mid z_t, X)$ evolves with the equation $\alpha_{t+1} = \alpha_t M_t(z_t; X)$, starting from $\alpha_1 = \mathrm{LM}_\theta^{(0)}(\cdot \mid X)$ until $\alpha_m$. Here, $M_t(z_t; X) \in \mathbb{R}^{v \times v}$ is a Markov transition matrix that is allowed to depend on the input $X$ and the last token $z_t$ (see (Ildiz et al., 2024) for related discussion). To keep exposition cleaner, we omit $z_t$ and $X$ in the notation of $M_t(z_t; X)$, and use $M_t$ instead. We first observe the following regarding base CoT2, standard discrete CoT, and CoT2-MTS inference strategies.

- *Base CoT2*: At each step $t = 1, \ldots, m$, the model outputs the continuous token $z_t = E^\top \alpha_t$ and uses it as the query for the next step.
  **Interpretation:** Base CoT2 keeps track of all possible traces simultaneously. Over $m$ steps, it tracks and aggregates all $v^m$ traces where the trace $(i_t)_{t=1}^m$ has a weight of $\prod_{i=1}^m \alpha_{t,i_t}$.

- *Discrete CoT*: At each step $1 \leq t \leq m$, the model samples exactly one token $z_t = e_{i_t}$ from $\alpha_t$, and uses it as the query

for the next step.

**Interpretation:** Discrete CoT samples a single trace out of $v^m$ traces with a likelihood of $\prod_{i=1}^{m} \alpha_{t,i_t}$ for trace $(i_t)_{t=1}^{m}$.

- **CoT2-MTS (multi-token sampling)**: At each step $1 \le t \le m$, i.i.d. sample $K$ tokens $\boldsymbol{e}_{i_1}, \ldots, \boldsymbol{e}_{i_K}$ from to $\boldsymbol{\alpha}_t$, average these tokens to form $\boldsymbol{z}_t = \frac{1}{K} \sum_{r=1}^{K} \boldsymbol{e}_{i_r}$, which it uses as query for the next step.

   **Interpretation:** CoT2-MTS tracks $K$ traces in parallel according to their discrete CoT likelihoods. However, these traces are not statistically independent.

With these three inference methods defined, we present the following result on the statistical consistency of the final outputs of these methods.

**Proposition 2** (Consistency of CoT and CoT2 inference). *Under Assumption 1 and given $\boldsymbol{X}$, the output of base CoT2 is $\boldsymbol{z}_m = \sum_{j=1}^{v} \alpha_{m,j} \boldsymbol{e}_j$ where $\boldsymbol{\alpha}_m = \boldsymbol{\alpha}_1 \prod_{t=1}^{m-1} \boldsymbol{M}_t$. Discrete CoT and CoT2-MTS have the same output once we take the expectation over their stochastic sampling.*

**Remark:** Please note that $\boldsymbol{\alpha}_m$ represents a distribution over the vocabulary. The proposition above shows that as the number of samples approaches infinity, the empirical distribution $\hat{\boldsymbol{\alpha}}_m$ obtained from CoT2-MTS (or discrete CoT) traces converge in probability to the deterministic distribution $\boldsymbol{\alpha}_m$ of the base CoT2 model. $\boldsymbol{\alpha}_m$ is computed in a sampling-free fashion and thus, is not a random variable.

Overall Proposition 2 establishes the statistical consistency of all three inference methods as they all estimate the same distribution $\boldsymbol{\alpha}_m$ over the vocabulary. However, they differ in how many samples are needed to approximate that distribution with repeated samplings. In particular, the base CoT2 model outputs the entire probability distribution over tokens at every intermediate step, thereby, implicitly tracking all possible trajectories in parallel as continuous embeddings. Consequently, it directly computes the exact final token distribution in one forward pass without repeated sampling. In contrast, due to stochasticity, discrete CoT or CoT2-MTS needs multiple i.i.d. samples to approximate this distribution. This observation motivates us to study and contrast the sample complexities of discrete CoT and CoT2-MTS. Our next proposition establishes our distribution approximation guarantee with respect to the total variation distance and shows that CoT2-MTS reduces the sample complexity of estimation compared to discrete CoT by a factor of $K$.

**Proposition 3.** *Let $\boldsymbol{\alpha}_m$ be the final expected output distribution after $m$ steps of CoT according to Proposition 2. Let $\hat{\boldsymbol{\alpha}}_m$ be the distribution resulting from averaging the outputs of i.i.d. CoT2-MTS traces with parallelism $K$. Then, to guarantee $\|\hat{\boldsymbol{\alpha}}_m - \boldsymbol{\alpha}_m\|_1 \le \epsilon$ with high probability, the total number of samples (traces) required scales as $\Theta\left(\frac{v}{K\epsilon^2}\right)$.*

Recall that CoT2-MTS generalizes discrete CoT ($K = 1$).

For $K = 1$, which corresponds to the discrete CoT model, the above proposition reduces to the known $\Theta(\frac{v}{\epsilon^2})$ sample complexity of approximating a $v$-category distribution in $\ell^1$ distance (Kamath et al., 2015). Note that as $K \to \infty$, CoT2-MTS converges to the base CoT2, and the above proposition recovers the one-shot performance of base CoT2. Thus, although the three models yield the same final distribution, the discrete model requires substantially more rollouts for accurate approximation due to inherent noise from single-token sampling. In contrast, the base CoT2 model carries the entire mixture of partial expansions at each step and computes the distribution in one shot, while the CoT2-MTS model captures multiple partial expansions in each step and proportionally reduces the sample complexity. This theoretical intuition aligns with our empirical findings in the Pass@k experiments in Section 3, where we observe that the base CoT2 approach achieves comparable performance to discrete CoT while requiring substantially fewer samples.

## 6. Limitations

One limitation of CoT2 is that it may require larger embedding dimensions to represent multiple parallel reasoning traces in more complex tasks. As a broader impact, while CoT2 can increase performance by searching more reasoning paths in parallel, this representation shadows the model's intermediate decision process, and might potentially reduce the interpretability of the model.

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

# APPENDIX

We discuss additional related work in Appendix A. We provide further implementation details in Appendix B, including those for the MNNS task (Appendix B.1), the ProntoQA/ProsQA datasets (Appendix B.2), and GRPO training (Appendix B.3). We present additional experimental results in Appendix C, and we offer details on continuous supervised training and GRPO in Appendix C.1 and Appendix C.2, respectively. Finally, we include the proofs of Propositions 1, 2, and 3 in Appendix D.

## A. Further Related Work

The proposed CoT2 approach simultaneously tracks all possible trajectories and superposes them within continuous tokens. This approach is similar to that of Xiong et al. (2024), who superpose multiple candidate outputs into a single final token. Our approach also shares similarities with decoding algorithms like self-consistency (Wang et al., 2022) and Best-of-N-Sampling (Stiennon et al., 2022), which generate multiple trajectories by running inference multiple times and then select a final answer based on the aggregate statistics. In contrast, our algorithm performs a single inference, superposing different trajectories all at once and determining the final answer in one pass. Furthermore, our Dirichlet sampling approach for generating multiple rollouts in GRPO training draws connections to previous works such as Latent Dirichlet Allocation (LDA) (Blei et al., 2003), which introduces Dirichlet priors within a hierarchical Bayesian framework, and AlphaGo (Silver et al., 2017), which injects Dirichlet noise to encourage exploration.

Our work also tangentially relates to research on multi-token prediction (Bachmann & Nagarajan, 2024; Liu et al., 2024; Gloeckle et al., 2024), which aims to improve the efficiency and quality of generation by predicting multiple tokens at once. It is hypothesized that effective future prediction necessitates the exploration of many possible continuations, which is similar to our CoT2 approach.

## B. Implementation Details

**Computational Resources:** All experiments were run on a Slurm-managed cluster using L40S GPUs with 48GB of memory. Each experiment fits on a single GPU. In the case of 4 input digits, the SFT or CSFT training takes approximately 3 hours on a single GPU. For 5-digit inputs, the dataset size increases by roughly a factor of 10, and the training time increases proportionally. The entire codebase was implemented in PyTorch.

### B.1. Implementation Details of Experiments on MNNS Task

**Dataset Details:** For the MNNS task, the vocabulary consists of a range of numbers from $[-S, S]$ for some positive integer $S$, together with *<BOS>*, *<EOS>*, and $\rightarrow$ special tokens. The integer $S$ is chosen so that all possible partial sums of the selected input digits lie within $[-S, S]$. For example, when the input digits lie in the range 1–10, we set $S = 36$, whereas for digits in 5–14, we set $S = 40$. We performed our experiments on the 4 and 5 input digit scenarios. A sample input line with $m$ numbers is:

$$<BOS> D_1 D_2 \ldots D_m \rightarrow$$

Accordingly, the output will be $m$ sum tokens, where the final token corresponds to the answer, followed by *<EOS>* token:

$$S_1 S_2 \ldots S_m <EOS>$$

As a concrete example, consider the input $2, 1, 4$ ($m = 3$), following Figure 1. In this case, the solution for the MNNS task is $-2 - 1 + 4 = 1$. Therefore, for the discrete model, the input is *<BOS>* $D_2 D_1 D_4 \rightarrow$ and we supervise it along the trajectory of correct output tokens $S_{-2} S_{-3}, S_1<EOS>$, as illustrated in Figure 1. On the other hand, the continuous supervision at the first step holds $S_2$ and $S_{-2}$ as possibilities. Then, for the next step, we add 1 or -1 to these numbers, and the resulting possibilities are $S_3, S_1, S_{-1}, S_{-3}$. Finally, at the last step, the model is supervised to pick the correct answer $S_1$ as the token.

We split the datasets by ensuring that each permutation of a set of numbers is exactly in one of the train and validation datasets, as the answer to the question is permutation-invariant. This way, we prevent the models from memorizing the answer and make a fair comparison. We also use 0.8-0.2 split for train-val datasets.

**Model and Hyperparameters:** We use the GPT2 model, with 1 layer 1 head, 2 layer 2 head, and 4 layer 4 head as the configurations. For each configuration, we experiment with embedding dimensions of 16, 24, or 32. We train with a learning rate of $\mathtt{lr} = 10^{-4}$ and use AdamW (no weight decay). The batch size is 16 for 4-digit inputs and 64 for 5-digit inputs.

**Evaluation of the models:** To make a proper comparison, we only check the final answer of the models, as checking the correctness of the full path of the discrete model would be unfair.

**Pass@k Experiment:** We perform our experiments for temperatures 0, 0.4, 0.8, and 1 by repeating the evaluation 10 times for each $k$ value where k changes from 1 to 14.

## B.2. Implementation Details of Experiments on ProntoQA/ProsQA Datasets

**Dataset Details:** Different from the original ProntoQA/ProsQA datasets which described the structured ontology in natural language as a set of known conditions, we use a more structured format through a token-level representation. An example prompt is shown below.

**Description of the structured ontology:** Each component of the ontology and associated questions is represented through discrete tokens with their own learned embeddings, rather than as raw textual input. Specifically, we use the GPT-2 architecture and encode the ontology's structural components. Below are two examples demonstrating how natural-language assertions are mapped to our tokenized format:

```
Brimpuses are not luminous → 'A' 'not in' 'B' '.'.
```

```
Shumpuses are amenable; Each yumpus is a lorpu; Every lorpus is floral → 'C' 'in' 'D' '.'.
```

Below, we have the ProntoQA and ProsQA datasets' input-output format.

**The structure of ProntoQA:**

**Input:**`'Description' '{' 'A' 'not in' 'B' '.' ... 'C' 'in' 'D' '.' '}' 'Question' '{' 'C' 'not in' 'F' '.' '}'`

**Output:**`'Steps' { 'C' 'in' 'D' '.' ... 'D' 'in' 'E' '.' '}' 'Answer' '{' 'False' '}'`

**The structure of ProsQA:**

**Input:**`'Description' '{' 'A' 'in' 'B' '.' ... 'C' 'in' 'D' '.' '}' 'Question' '{' 'C' 'in' 'F' 'or' 'E' '}'`

**Output:**`'Steps' { 'C' 'in' 'D' '.' ... 'D' in 'E' '.' '}' 'Answer' '{' 'F' '}'`

Each distinct component or relation (e.g., `'A'`, `'in'`, `'not in'`) is treated as a unique token, and singular/plural variants (such as `'lempus'` and `'lempuses'`) are collapsed into a single token to simplify the vocabulary. Alongside these concept tokens, special structural tokens (`'Description'`, `''`, `''`,`'.'`, `'or'`, etc.) are also included, which results in a vocabulary size of 31 tokens. To avoid biases, we balance the dataset. In ProntoQA, "yes" and "no" each appear with 50% probability, and in ProsQA, the correct answer is randomly permuted at the first or second position. For all the other experimental and training settings, we follow (Hao et al., 2024).

**Model and Hyperparameters:** We use the GPT2 model, with 2 layer 2 head, and 4 layer 4 head as the configurations. We tested embedding dimensions 24, 32, 40 with these configurations. We set batch size 64. We train with a learning rate of $10^{-4}$ and use AdamW (no weight decay).

**Maj@k Experiment:** We use majority voting for evaluation instead of Pass@k, because both ProntoQA and ProsQA are binary questions. We perform our experiments for temperatures 0, 0.4, 0.8, and 1 by repeating the evaluation 10 times for each $k$ value where k changes from 1 to 21. If two or more answers end up with the same top vote, we pick one randomly.

## B.3. Implementation Details of GRPO Training

In (Hao et al., 2024), the reference model is updated by $\text{LM}_{\theta_{\text{ref}}} \leftarrow \text{LM}_\theta$ in each iteration (epoch). This approach is reasonable for their setting with a large dataset and a small number of epochs over it. For our setting, however, we set the reference model to the initial model and never update it through iterations as we have a smaller dataset. Meanwhile, we update the old model before every batch $\text{LM}_{\theta_{\text{old}}} \leftarrow \text{LM}_\theta$.

In our experiments, we use $G = 8$ trajectories per input data point, use clipping parameter $\epsilon = 0.1$, and set the KL-divergence coefficient $\beta = 0$ in most cases (with $\beta = 0.1$ in a few). For the CoT2 model with MTS sampling, we change the number of tokens to sample $K$ from 1 to 12. In the MNNS task, the 5-digit case has about ten times more data than the 4-digit

case, so we typically focus on 4-digit MNNS because of computational considerations and use a batch size of 16 in those experiments.

Learning rates differ by model and setting. We use $\texttt{lr} = 5 \times 10^{-5}$ for CoT2-MTS sampling (figures in the main text), $\texttt{lr} = 1 \times 10^{-5}$ for discrete CoT with Dirichlet sampling, and $\texttt{lr} = 1 \times 10^{-6}$ for CoT2 with Dirichlet sampling. For ProntoQA and ProsQA experiments, we perform a grid search over learning rates ranging from $1 \times 10^{-4}$ to $1 \times 10^{-8}$ and select and report results using the best-performing configuration. For most settings, we find $\texttt{lr} = 1 \times 10^{-5}$ optimal; however, for CoT2 and discrete CoT models with $K = 6$, we set $\texttt{lr} = 1 \times 10^{-6}$. We also use AdamW with a weight-decay of 0.01. For Dirichlet experiments on the MNNS task, we try various scale parameters $\gamma$, but we find $\gamma = 20$ to work best in most settings. Unless stated otherwise, we report the best validation accuracy found during training for each setting.

## C. Experimental Results

### C.1. Continuous Supervised Training Results

**Teacher Forcing and Self-feeding Comparison:** As described in Section 3, we tested two approaches of providing prefixes during training the CoT2 model with CSFT. Although the model autoregressively generates at inference time, teacher forcing yields better performance than self-feeding during CSFT training. Our results demonstrate that, We also tested curriculum settings, where we switch to self-feeding after a pre-determined number of epochs in the training. Still, the accuracies didn't improve beyond pure teacher-forcing training. The results are illustrated in Figure 4, where we refer to teacher-forcing as "hard-teacher" and refer to self-feeding as "soft-teacher".

**Sparse Supervision for Discrete Baseline:** We also tested providing a subset of the correct path to the discrete model. We observed that a sparsely supervised discrete model can achieve better performance than the fully supervised discrete model when the distribution is "easier" to handle by the model. As an example, we tested the case when we have 5 input digits from the range of 11 to 19. In this case, in nearly all of the cases, the answer to our MNNS game is (sum of minimum 3 numbers) - (sum of maximum 2 numbers) out of the 5 input numbers. In this case, when only 1 token from the correct path is provided to the discrete model, it's better than 3 and 5 token cases. However, when we change the distribution to a range of numbers from 5 to 13, which makes the question reasonably harder, the discrete model with 1 token supervision performs worse than the other two, and the discrete model with full supervision performs best. The results are demonstrated in Figure 3.

**Further Results on CoT2 vs Discrete CoT:** The results in Figure 5 also indicate that above an embedding dimension threshold, the CoT2 model has superior performance and trains significantly faster than the discrete CoT model. Moreover, combining the results of Figure 5 with Figure 2c, we see that the CoT2 model with one layer and one head GPT2 model performs better than discrete CoT model with two layers and two heads at embeddings 24 and 32. While the continuous approach requires greater embedding capacity to support its distributional representations at each step, it can outperform the discrete model using fewer layers and attention heads. Supporting the findings in Figure 2, Figure 6 illustrates that on the ProntoQA task, CoT2 consistently outperforms the discrete CoT baseline when the embedding dimension is above a threshold. Likewise, as depicted in Figure 7 and Figure 8, the discrete CoT model requires multiple samplings (Maj@k) to match the single-shot performance of CoT2 on both ProntoQA and ProsQA, which indicates that CoT2 model is more sample-efficient.

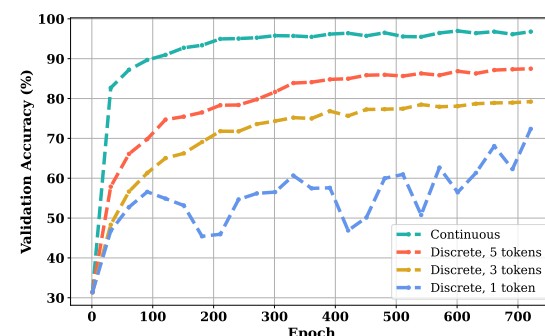

Figure 3: The figure illustrates that when the range of digits makes the question non-trivial on an MNNS task, the discrete CoT model trained with full token supervision outperforms sparse supervisions; in particular, single token supervision yields the worst performance. **Setting:** 5 input digits in $5 - 13$; 2-layer, 2-head GPT2 with $d = 32$.

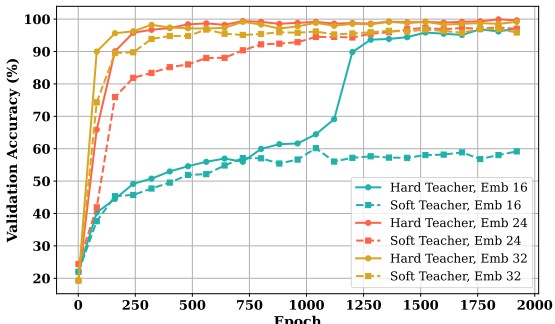

Figure 4: The comparison between the hard and soft teachers for different embedding dimensions. The figure illustrates that the hard teacher is superior to the soft teacher. **Setting:** 4 input digits in $1 - 9$; 4-layer, 4-head GPT2 with $d \in \{16, 24, 32\}$.

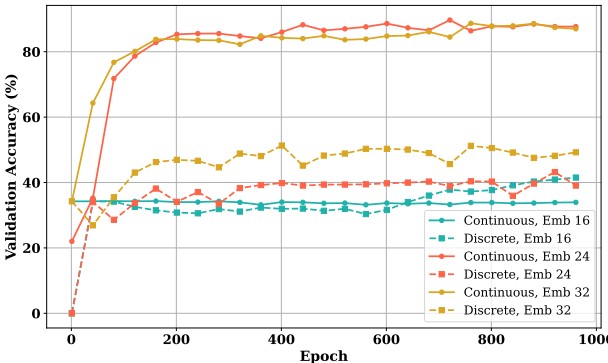

Figure 5: Comparison between CoT2 and discrete CoT2 model for different embedding dimensions. The figure demonstrates that above a certain embedding dimension threshold, the CoT2 model outperforms the discrete CoT model in the MNNS task. **Setting:** 4 input digits in $1 - 9$; 1-layer, 1-head GPT2 with $d \in \{16, 24, 32\}$.

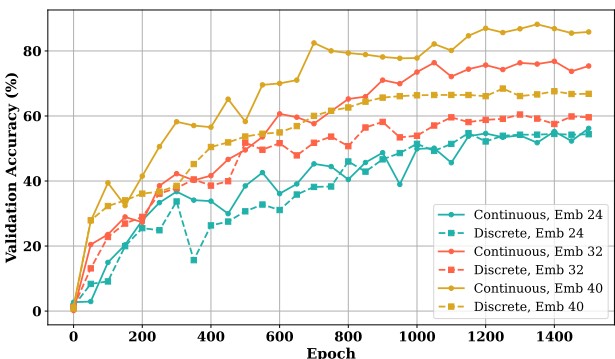

Figure 6: Comparison between CoT2 and discrete CoT2 model for different embedding dimensions in ProntoQA task. The figure shows that above an embedding dimension threshold, the CoT2 model outperforms the discrete CoT model. **Setting:** 4 input digits in $1 - 9$; 4-layer, 4-head GPT2 with $d \in \{24, 32, 30\}$.

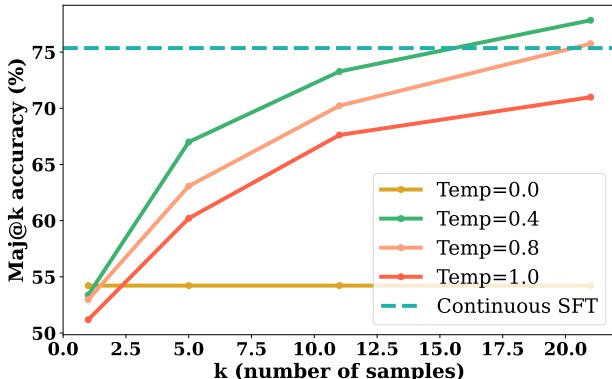

Figure 7: The figure illustrates that the discrete CoT2 model requires multiple samplings (Maj@k) to match the single-shot performance of the CoT2 model on ProntoQA (10-run average). **Setting:** 4-layer, 4-head GPT2 with $d = 32$.

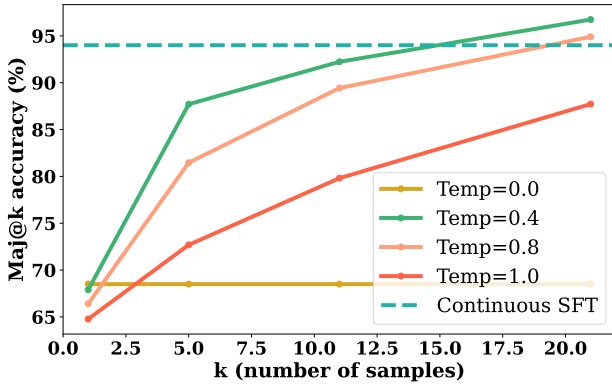

Figure 8: The figure illustrates that the discrete CoT2 model requires multiple samplings (Maj@k) to match the single-shot performance of the CoT2 model on ProsQA (10-run average). **Setting:** 4-layer, 4-head GPT2 with $d = 32$.

## C.2. GRPO Results

**Discussion on ProntoQA/ProsQA Datasets:** Table 2 illustrates that GRPO training using CoT2-MTS sampling consistently improves discrete CoT and CoT2 models over their initial SFT accuracy. Moreover, we observe that the improvement in the discrete CoT model is greater, which might indicate that the CoT2 model already gains an RL-like exploration mechanism through CSFT training. We observe that while increasing $K$ initially increases the accuracy by sampling more tokens at each step, beyond some $K$, the improvements diminish. This observation is consistent with Table 1, where we see that a moderate $K$ value offers the best final performance. One possible explanation is that while higher $K$ promotes better exploration, it also raises the chance of sampling unhelpful tokens that disrupt the averaged token representation. Indeed, for larger $K$, we observe that the RL objective saturates to near zero which suggests that most rollouts fail once the averaged token contains too many distracting tokens.

**Discussion on Dirichlet Sampling:** We also investigate the effects of Dirichlet sampling in GRPO training discrete CoT and CoT2 models. The results in Table 3 indicates that applying Dirichlet sampling ($\gamma = 20$) in GRPO training of discrete CoT model consistently improves over the initial SFT training accuracies. Similar to the CoT2 + MTS sampling results in Table 1, we observe that the entropy at the third token remains relatively high, which suggests a beneficial diversity in model's predictions for that token. Moreover, the Table 4 demonstrates that Dirichlet sampling also improves the CoT2 model's SFT accuracy, even though it has a high initial SFT accuracy. As illustrated in Table 4, we find there is an optimal value for the scale parameter $\gamma$, since larger $\gamma$ typically yields more uniform sampling distributions, whereas smaller $\gamma$ concentrates the distribution more sharply. Thus, adjusting $\gamma$ provides a balance between exploration and stability in GRPO training.

Table 3: Discrete CoT models trained with GRPO after SFT using Dirichlet sampling ($\gamma = 20$) and a learning rate of $1 \times 10^{-5}$. We show validation accuracy (%) and token-level entropy (SFT $\rightarrow$ SFT+GRPO) for each (Layers, Heads) setting, with an embedding dimension of 24 for GPT2 model.

| Layers | Heads | Val. Accuracy (%) | | Val. Entropy (SFT $\rightarrow$ SFT+GRPO) | | | |
| --- | --- | --- | --- | --- | --- | --- | --- |
| | | SFT | SFT+GRPO | $token_1$ | $token_2$ | $token_3$ | $token_4$ |
| 1 | 1 | 39.76 | 46.25 | $0.4851 \rightarrow 0.1701$ | $0.5165 \rightarrow 0.6380$ | $0.3243 \rightarrow 0.6590$ | $0.1597 \rightarrow 0.4878$ |
| 2 | 2 | 70.26 | 75.84 | $0.4851 \rightarrow 0.4027$ | $0.5165 \rightarrow 0.4413$ | $0.3243 \rightarrow 0.2907$ | $0.1597 \rightarrow 0.1386$ |

Table 4: Validation accuracies GRPO training with CoT2 models using different Dirichlet sampling scales ($\gamma$) with learning rate of $1 \times 10^{-6}$. We show the baseline SFT accuracy (87.84%) and final performance after GRPO.

| Dirichlet Scale ($\gamma$) | SFT Val. Acc (%) | SFT + GRPO Val. Acc (%) |
| --- | --- | --- |
| 10 | | 89.76 |
| 20 | 87.84 | 90.75 |
| 40 | | 90.37 |

# D. Theoretical Details

**Justification for Assumption 1:** The Assumption 1 holds for tasks where the next-token distribution depends solely on the current token and the input tokens rather than the full history of output tokens. This is satisfied by many reasoning tasks, where the aim is to keep track of an intermediate state (e.g., the current sum) and update this state based only on the current state and the input, independently of the earlier trajectory.

For example, in the MNNS task, the model generates a token representing the current partial sum at each step. To compute the distribution over the next possible sums, the model adds or subtracts the selected number from the input context $X$ to the current sum, without needing to remember the sequence of previous sums explicitly. Thus, the next-state distribution at each step is only determined by the current state and it naturally satisfies the Assumption 1.

**Proposition 2** (Consistency of CoT and CoT2 inference)**.** *Under Assumption 1 and given $X$, the output of base CoT2 is $z_m = \sum_{j=1}^{v} \alpha_{m,j} e_j$ where $\alpha_m = \alpha_1 \prod_{t=1}^{m-1} M_t$. Discrete CoT and CoT2-MTS have the same output once we take the expectation over their stochastic sampling.*

*Proof.* Let $\hat{\alpha}_t^{(\text{disc})}, \hat{\alpha}_t^{(\text{MTS})}$ denote the empirical output token distributions at step $t$ under one trajectory obtained by the discrete CoT, and CoT2 with MTS models, respectively. We define $\alpha_t^{(\text{disc})} = \mathbb{E}\left[\hat{\alpha}_t^{(\text{disc})}\right]$ and $\alpha_t^{(\text{MTS})} = \mathbb{E}\left[\hat{\alpha}_t^{(\text{MTS})}\right]$ to be corresponding expected distributions. The discrete CoT model at each step picks exactly 1 token from $\alpha_t$. On the other hand, CoT2-MTS samples $K$ i.i.d. tokens at every step independently according to their probabilities from $\hat{\alpha}_t$. We denote them $i_1, \ldots, i_K$, and average their embeddings to produce a single query.

We will use induction in our argument. For the base case, all models start with the same initial distribution, so we trivially have $\alpha_1^{(\text{disc})} = \alpha_1^{(\text{MTS})} = \alpha_1^{(\text{CoT2})}$. For the inductive step, assume that we have $\alpha_{t-1}^{(\text{disc})} = \alpha_{t-1}^{(\text{MTS})} = \alpha_{t-1}^{(\text{CoT2})}$. We will show that $\alpha_t^{(\text{disc})} = \alpha_t^{(\text{MTS})} = \alpha_t^{(\text{CoT2})}$. On the other, for the discrete CoT model, the model samples one token $e_{i_{t+1}}$ from the row of $M_t$ for a token $i_{t+1}$. Therefore, we need to condition on the token at step $t$. We have:

$$\mathbb{E}\left[\hat{\alpha}_{t+1}^{(\text{disc})}\right] = \sum_{j=1}^{v} \mathbb{P}(z_t = e_j)\, \mathbb{E}\left[\hat{\alpha}_{t+1}^{(\text{disc})} \mid z_t = e_j\right] = \sum_{j=1}^{v} \mathbb{P}(z_t = e_j)\, \text{LM}_\theta^{(t)}(\cdot \mid e_j, X)$$

$$\sum_{j=1}^{v} \alpha_{t,j}^{(\text{disc})}\, \text{LM}_\theta^{(t)}(\cdot \mid e_j, X)$$

$$\overset{(a)}{=} \sum_{j=1}^{v} \alpha_{t,j}^{(\text{CoT2})}\, \text{LM}_\theta^{(t)}(\cdot \mid e_j, X)$$

$$\overset{(b)}{=} \alpha_{t+1}^{(\text{CoT2})} \tag{6}$$

where (a) follows from the induction argument and (b) follows from Assumption 1. Therefore, we obtain $\alpha_{t+1}^{(\text{disc})} = \mathbb{E}\left[\hat{\alpha}_{t+1}^{(\text{disc})}\right] = \alpha_{t+1}^{(\text{CoT2})}$. For the CoT2-MTS model, the argument will be similar. Using the decoupling of trajectories by Assumption 1, the next distribution is:

$$\text{LM}_\theta^{(t)}\left(\cdot \mid \frac{1}{K}\sum_{r=1}^{K} e_{i_r}, X\right) = \frac{1}{K}\sum_{r=1}^{K} \text{LM}_\theta^{(t)}\left(\cdot \mid e_{i_r}, X\right).$$

Therefore, we write:

$$\mathbb{E}\left[\hat{\boldsymbol{\alpha}}_{t+1}^{(\text{MTS})}\right] = \sum_{(i_1,\ldots,i_K)\in[v]^K} \mathbb{P}(\boldsymbol{e}_{i_1},\ldots,\boldsymbol{e}_{i_K})\,\mathbb{E}\left[\hat{\boldsymbol{\alpha}}_{t+1}^{(\text{MTS})} \mid \boldsymbol{e}_{i_1},\ldots,\boldsymbol{e}_{i_K}\right]$$

$$= \sum_{(i_1,\ldots,i_K)\in[v]^K} \mathbb{P}(\boldsymbol{e}_{i_1},\ldots,\boldsymbol{e}_{i_K})\,\text{LM}_\theta^{(t)}\left(\cdot \mid \frac{1}{K}\sum_{r=1}^{K}\boldsymbol{e}_{i_r},\boldsymbol{X}\right)$$

$$= \sum_{(i_1,\ldots,i_K)\in[v]^K} \mathbb{P}(\boldsymbol{e}_{i_1},\ldots,\boldsymbol{e}_{i_K})\frac{1}{K}\sum_{r=1}^{K}\text{LM}_\theta^{(t)}\left(\cdot \mid \boldsymbol{e}_{i_r},\boldsymbol{X}\right)$$

$$= \sum_{(i_1,\ldots,i_K)\in[v]^K} \left(\prod_{r=1}^{K}\mathbb{P}(\boldsymbol{e}_{i_r})\right)\frac{1}{K}\sum_{r=1}^{K}\text{LM}_\theta^{(t)}\left(\cdot \mid \boldsymbol{e}_{i_r},\boldsymbol{X}\right)$$

$$= \sum_{(i_1,\ldots,i_K)\in[v]^K} \left(\prod_{r=1}^{K}\alpha_{t,i_r}^{(\text{MTS})}\right)\frac{1}{K}\sum_{r=1}^{K}\text{LM}_\theta^{(t)}\left(\cdot \mid \boldsymbol{e}_{i_r},\boldsymbol{X}\right)$$

$$= \frac{1}{K}\sum_{r=1}^{K}\sum_{j=1}^{v}\text{LM}_\theta^{(t)}(\cdot \mid \boldsymbol{e}_j,\boldsymbol{X})\sum_{(i_1,\ldots,i_{r-1},i_{r+1},\ldots,i_K)\in[v]^{K-1}}\alpha_{t,j}^{(\text{MTS})}\prod_{\substack{s=1\\s\neq r}}^{K}\alpha_{t,i_s}^{(\text{MTS})}$$

$$= \frac{1}{K}\sum_{r=1}^{K}\sum_{j=1}^{v}\alpha_{t,j}^{(\text{MTS})}\,\text{LM}_\theta^{(t)}(\cdot \mid \boldsymbol{e}_j,\boldsymbol{X})\sum_{(i_1,\ldots,i_{r-1},i_{r+1},\ldots,i_K)\in[v]^{K-1}}\prod_{\substack{s=1\\s\neq r}}^{K}\alpha_{t,i_s}^{(\text{MTS})}$$

$$= \sum_{r=1}^{K}\frac{1}{K}\sum_{j=1}^{v}\alpha_{t,j}^{(\text{MTS})}\,\text{LM}_\theta^{(t)}\left(\cdot \mid \boldsymbol{e}_j,\boldsymbol{X}\right) = \sum_{j=1}^{v}\alpha_{t,j}^{(\text{MTS})}\,\text{LM}_\theta^{(t)}\left(\cdot \mid \boldsymbol{e}_j,\boldsymbol{X}\right)$$

$$= \sum_{j=1}^{v}\alpha_{t,j}^{(\text{CoT2})}\,\text{LM}_\theta^{(t)}\left(\cdot \mid \boldsymbol{e}_j,\boldsymbol{X}\right) = \boldsymbol{\alpha}_{t+1}^{(\text{CoT2})}. \tag{7}$$

Thus, combining (6), and (7) completes the induction and our argument:

$$\boldsymbol{\alpha}_{t+1}^{(\text{disc})} = \mathbb{E}\left[\hat{\boldsymbol{\alpha}}_{t+1}^{(\text{disc})}\right] = \boldsymbol{\alpha}_{t+1}^{(\text{CoT2})} = \mathbb{E}\left[\hat{\boldsymbol{\alpha}}_{t+1}^{(\text{MTS})}\right] = \boldsymbol{\alpha}_{t+1}^{(\text{MTS})}.$$

$\square$

**Proposition 3.** *Let $\boldsymbol{\alpha}_m$ be the final expected output distribution after $m$ steps of CoT according to Proposition 2. Let $\hat{\boldsymbol{\alpha}}_m$ be the distribution resulting from averaging the outputs of i.i.d. CoT2-MTS traces with parallelism $K$. Then, to guarantee $\|\hat{\boldsymbol{\alpha}}_m - \boldsymbol{\alpha}_m\|_1 \leq \epsilon$ with high probability, the total number of samples (traces) required scales as $\Theta\left(\frac{v}{K\epsilon^2}\right)$.*

*Proof.* We will utilize the empirical distributions $\hat{\boldsymbol{\alpha}}_t^{(\text{disc})}, \hat{\boldsymbol{\alpha}}_t^{(\text{MTS})}$ that are defined in the proof of Proposition 3 and show that i.i.d. sampling $K$ discrete CoT trajectories and averaging their results at the last step is distributionally equivalent to CoT2-MTS using $K$ tokens, under Assumption 1. We will first argue the results for $m = 2$ and then we will show it for any $m = t + 1$. As discussed in the previous proposition, using the decoupling of trajectories by Assumption 1, the next distribution $\hat{\boldsymbol{\alpha}}_1^{(\text{MTS})}$ when $\boldsymbol{e}_{i_1},\ldots,\boldsymbol{e}_{i_K}$ are drawn from $\boldsymbol{\alpha}_1$ is:

$$\hat{\boldsymbol{\alpha}}_2^{(\text{MTS})} = \text{LM}_\theta^{(1)}\left(\cdot \mid \frac{1}{K}\sum_{r=1}^{K}\boldsymbol{e}_{i_r},\boldsymbol{X}\right)$$

$$= \frac{1}{K}\sum_{r=1}^{K}\text{LM}_\theta^{(1)}\left(\cdot \mid \boldsymbol{e}_{i_r},\boldsymbol{X}\right)$$

$$= \frac{\sum_{r=1}^{K}\text{LM}_\theta^{(1)}\left(\cdot \mid \boldsymbol{e}_{i_r},\boldsymbol{X}\right)}{K}$$

$$= \frac{\sum_{r=1}^{K}\hat{\boldsymbol{\alpha}}_{2,r}^{(\text{disc})}}{K}, \tag{8}$$

which is the empirical mean of $K$ i.i.d. discrete CoT draws. Thus, under our assumption, drawing $K$ tokens with the MTS approach is distributionally equivalent to combining outcomes of $K$ independent discrete CoT draws. Now, for any $t$, we know by Assumption 1 that output at step $t + 1$ does not depend on the previous history of tokens. Because of this, we only focus on the tokens $e_{i_1}, \ldots, e_{i_K}$ drawn at step $t$ from $\alpha_t^{(\text{MTS})}$, where $\alpha_t^{(\text{MTS})} = \alpha_t^{(\text{disc})}$ according to Proposition 2. Following the same steps as in (8), we obtain that:

$$\hat{\alpha}_{t+1}^{(\text{MTS})} = \frac{\sum_{r=1}^{K} \text{LM}_\theta^{(t)} (\cdot \mid e_{i_r}, X)}{K} = \frac{\sum_{r=1}^{K} \hat{\alpha}_{t+1,r}^{(\text{disc})}}{K},$$

which is again the empirical mean of $K$ i.i.d. discrete CoT trajectory output token distributions. To finish our argument, we will benefit from a standard result in multinomial estimation that $\Theta\left(\frac{v}{\epsilon^2}\right)$ i.i.d. samples are necessary and sufficient to learn a $v$-category distribution in $\| \cdot \|_1$-distance $\leq \epsilon$ (Kamath et al., 2015). In our MTS setting, each sample uses $K$ i.i.d. draws internally. This reduces the variance by a factor of $K$ and achieves the same estimation guarantee with $\Theta\left(\frac{v}{K\epsilon^2}\right)$ aggregated samples. Hence the total sample complexity in terms of MTS samplings is $\Theta\left(\frac{v}{K\epsilon^2}\right)$, as claimed. This completes the argument. $\qquad \square$

### D.1. Construction for Minimum Non-Negative Sum (MNNS) Task

We describe a single-layer transformer with an attention block followed by a mixture-of-experts (MoE) feed-forward block. Let $n$ be the length of the input sequence of integer tokens. Denote the tokenized input numbers as $z_1, z_2, \ldots, z_n$; and let the arrow ($\rightarrow$) token be denoted as $z_{n+1}$. We also have a dummy input token $z_{n+2}$, which is the embedding corresponding to number 0, so that we have $n + 2$ tokens initially. We will construct the transformer with $n + 1$ MLPs in the mixture of experts layer, where the first $n$ are partial-sum MLPs and the last one is the MLP that reads off the answer from among the all stored partial sums after $m$ steps. We start with the following assumption on the structure of the tokens.

**Assumption 2.** *Let $d = d_e + d_p$ be the embedding size where $d_e = 2^{n+1}$ and $d_p = n + 2$. The token embeddings are on the first $d_e$ coordinates, while the positional encodings are on the last $d_p$ coordinates and are one-hot encoded. where each*
$$z_i = \begin{pmatrix} e_i \\ \mathbf{p}_i \end{pmatrix} \in \mathbb{R}^{d_e + d_p} \text{ is formed by vertically concatenating a content embedding } e_i \in \mathbb{R}^{d_e} \text{ and a positional encoding } \mathbf{p}_i \in \mathbb{R}^{d_p}.$$
*We assume each $\mathbf{p}_i$ is a one-hot vector in $\mathbb{R}^{d_p}$, so that $\mathbf{p}_i^\top \mathbf{p}_j = 0$ for $i \neq j$, and $\|\mathbf{p}_i\| = 1$.*

We now state the following proposition, which helps us to attend and select the input tokens $z_1, \ldots, z_{n+1}$ one by one by the attention block.

**Proposition 4.** *Suppose we have $n + 2$ tokens $\{z_1, z_2, \ldots, z_{n+2}\}$ in $\mathbb{R}^d$, each of the form $z_i = \begin{pmatrix} e_i \\ \mathbf{p}_i \end{pmatrix}$, where $e_i \in \mathbb{R}^{d_e}, \mathbf{p}_i \in \mathbb{R}^{d_p}, d = d_e + d_p$. Let $p_1, p_2, \ldots, p_{n+2} \in \mathbb{R}^{d_p}$ be orthonormal set of positional vectors according to Assumption 2. Then, there exists a rotation matrix $R \in \mathbb{R}^{d_p \times d_p}$ satisfying $Rp_j = p_{j-1 \bmod (n+2)}$ for all $j \in [n + 2]$, and the block matrices*

$$W = \begin{pmatrix} \mathbf{0}_{d_e \times d_e} & \mathbf{0}_{d_e \times d_p} \\ \mathbf{0}_{d_p \times d_e} & c \cdot R \end{pmatrix} \in \mathbb{R}^{d \times d} \quad \text{and} \quad W_v = \begin{pmatrix} I_{d_e} & \mathbf{0}_{d_e \times d_p} \\ \mathbf{0}_{d_p \times d_e} & I_{d_p} \end{pmatrix} \in \mathbb{R}^{d \times d}$$

*with $c \rightarrow \infty$, ensure that the attention block*

$$\text{Attn}(z, Z) = \mathbb{S}\left(z^\top W Z^\top\right) Z W_v,$$

*performs a cyclic next-index selection: if the query is $z_i$, it selects column $j^* \equiv (i + 1) \pmod{n + 2}$ from $Z$ and returns $z_{j^*}$.*

*Proof. Definition of Matrix $W$.* We will first construct a rotation matrix. We have $n + 2$ orthonormal position vectors $p_1, \ldots, p_{n+2} \in \mathbb{R}^{d_p}$. Then, $R$ is the following $(n + 2) \times (n + 2)$ permutation matrix

$$R = \begin{pmatrix} 0 & 1 & 0 & \cdots & 0 & 0 \\ 0 & 0 & 1 & \cdots & 0 & 0 \\ 0 & 0 & 0 & \cdots & 0 & 0 \\ \vdots & \vdots & \vdots & \ddots & \vdots & \vdots \\ 0 & 0 & 0 & \cdots & 0 & 1 \\ 0 & 0 & 0 & \cdots & 0 & 0 \end{pmatrix},$$

which cyclically shifts the basis vectors $p_j$ backward by one index, i.e., $Rp_j = p_{j-1 \bmod (n+2)}$. Then, we specify

$$W = \begin{pmatrix} \mathbf{0}_{d_e \times d_e} & \mathbf{0}_{d_e \times d_p} \\ \mathbf{0}_{d_p \times d_e} & c \cdot R \end{pmatrix} \in \mathbb{R}^{d \times d}.$$

Hence for $z_i = (e_i; p_i)$, we have $\left(e_i^\top, p_i^\top\right) W = \left(\mathbf{0}, p_i^\top R\right)$. Thus the dot-product with $z_j$ is

$$\begin{pmatrix} 0 & p_i^\top R \end{pmatrix} \begin{pmatrix} e_j \\ p_j \end{pmatrix} = p_i^\top R p_j.$$

Since positional encodings are orthogonal, we know that:

$$p_i^\top R p_j = \begin{cases} 1, & j \equiv i + 1 (\bmod(n + 2)), \\ 0, & \text{else.} \end{cases}$$

So row-wise softmax $\mathbb{S}\left(x^\top W X^\top\right)$ places all probability mass at column $j^* \equiv i + 1(\bmod(n + 2))$ by saturating softmax at position $j$ as $c \to \infty$.

*Definition of Matrix $W_v$.* In this case, we simply set $W_v = I_d$, and thus, once the row-wise softmax selects column $j^*$ with probability 1, we have

$$z_{j^*}^\top W_v = z_{j^*},$$

so the final output is precisely the chosen $z_{j^*}$. This completes the construction. $\qquad\square$

Having defined the attention block, we state the following proposition that helps selecting different MLPs for the tokens $z_1, \ldots, z_{n+1}$ outputted by the attention block.

Having defined the attention block, we now show how a mixture-of-experts layer can exclusively select $MLP_i$ for each token $z_i$, $i = 1, \ldots, n + 1$ outputted by the attention block.

**Proposition 5.** *Let $MLP_1, \ldots, MLP_{n+1}$ be $n + 1$ experts in a mixture-of-experts (MoE) module. Suppose we have $n + 1$ fixed token embeddings $\{z_1, z_2, \ldots, z_{n+1}\} \subset \mathbb{R}^d$, where each token is formed according to Assumption 2. Given routing parameters $W = [w_1 \ \ldots \ w_{n+1}]^\top$, define the MoE feed-forward block as*

$$\text{MoEBlock}(z) = \sum_{j=1}^{n+1} \left[ \text{Softmax}(Wz)_j \cdot MLP_j(z) \right],$$

*where*

$$\text{Softmax}(Wz)_j = \frac{\exp\left(w_j^\top z\right)}{\sum_{k=1}^{n+1} \exp\left(w_k^\top z\right)}, \quad j = 1, \ldots, n + 1.$$

*There exist routing matrix $W \in \mathbb{R}^{(n+1) \times d}$ such that the distribution $\text{Softmax}(c \cdot Wz_i)$ as $c \to \infty$ assigns a weight of 1 on $MLP_i$ when $z_i$ is given as input.*

*Proof.* We partition $w_j$ to ignore the content embedding $e_i$ and match the positional block $p_j$. Concretely, write $w_j = \begin{pmatrix} \mathbf{0}_{d_e} \\ p_j \end{pmatrix}$. Then, for each token $z_i = (e_i; p_i)$,

$$w_j^\top z_i = \begin{pmatrix} \mathbf{0}_{d_e}^\top & p_j^\top \end{pmatrix} \begin{pmatrix} e_i \\ p_i \end{pmatrix} = p_j^\top p_i.$$

Since $p_j^\top p_i = \delta_{ij}$, we have $w_j^\top z_i = \delta_{ij}$. Therefore, the softmax evaluates to

$$\lim_{c \to \infty} \text{Softmax}(c \cdot Wz_i)_j \to \frac{\exp(\delta_{ij})}{\sum_{k=1}^{n+1} \exp(\delta_{ik})} = \delta_{ij}.$$

In other words, $\text{Softmax}(c \cdot Wz_i)$ places all mass on expert $j = i$. Thus each token $z_i$ (for $i = 1, \ldots, n + 1$) deterministically selects the $i$-th expert $MLP_i$. $\qquad\square$

In the next proposition, we show how to iteratively expand the partial sums by adding and subtracting the digit obtained from the attention block and write each resulting sum to a distinct spot in the output vector.

**Proposition 6** (Partial-Sum MLPs)**.** *Suppose that the embedding dimension $d$ satisfies $d \geq 2^{j+1} + d_p$. Let $z_{prev}$ contain the $2^{j-1}$ partial sums $s_k$ each encoded by a pair $(\cos(\omega s_k), \sin(\omega s_k))$ of coordinates such that:*

$$z_{prev} = \begin{bmatrix} \cos(\omega s_1) \sin(\omega s_1) & \dots & \cos(\omega s_{2^{j-1}}) \sin(\omega s_{2^{j-1}}) \, 0 \dots 0 \end{bmatrix}^\top \in \mathbb{R}^d,$$

*and let $z_{curr}$ contain the input digit $d_j$ encoded in the first two coordinates:*

$$z_{curr} = \begin{bmatrix} \cos(\omega d_j) \sin(\omega d_j) \, 0 & \dots & 0 \end{bmatrix}^\top \in \mathbb{R}^d.$$

*Then, for any $1 \leq j \leq n$, there exist $MLP_j : \mathbb{R}^d \times \mathbb{R}^d \to \mathbb{R}^d$ such that when $(z_{prev}, z_{curr})$ is given as input, it outputs the vector $z_{out} \in \mathbb{R}^d$ so that its first $2^j$ coordinate-pairs store the trigonometric encodings of $(s_k + d_j)$, and the next $2^j$ coordinate-pairs store those of $(s_k - d_j)$. Formally, first $2^j$ coordinates are $[\cos(\omega(s_k + d_j)), \sin(\omega(s_k + d_j))]$ for all partial sums $s_k$, and the next $2^j$ coordinates are $[\cos(\omega(s_k - d_j)), \sin(\omega(s_k - d_j))]$ for all partial sums $s_k$, with any remaining coordinates set to zero.*

*Proof.* Each expert $MLP_j$ (for $1 \leq j \leq n$) adds $j$-th integer $d_j$ in both its positive and negative form to all previously computed partial sums. For simplicity, let's say that $j$-th integer to add is $d_j$. By trigonometric identities, we know that

$$\cos(\omega(s_k + d_j)) = \cos(\omega s_k)\cos(\omega d_j) - \sin(\omega s_k)\sin(\omega d_j),$$
$$\sin(\omega(s_k + d_j)) = \sin(\omega s_k)\cos(\omega d_j) + \cos(\omega s_k)\sin(\omega d_j),$$

and similarly,

$$\cos(\omega(s_k - d_j)) = \cos(\omega s_k)\cos(\omega d_j) + \sin(\omega s_k)\sin(\omega d_j),$$
$$\sin(\omega(s_k - d_j)) = \sin(\omega s_k)\cos(\omega d_j) - \cos(\omega s_k)\sin(\omega d_j).$$

Using the above identities, we will obtain the sum by introducing matrices that do shift/swap operations. Concretely, for $k = 1, \dots, 2^m$, the $k$-th $2 \times 2$ block acts on $\begin{pmatrix} \cos(\omega s_k) \\ \sin(\omega s_k) \end{pmatrix}$ in $z_{\text{prev}}$. We define:

$$W_{\text{sin}}^+ = \text{diag} \left( \underbrace{\begin{pmatrix} 0 & -1 \\ 1 & 0 \end{pmatrix}, \dots, \begin{pmatrix} 0 & -1 \\ 1 & 0 \end{pmatrix}}_{2^{j-1} \text{ blocks}}, 0, \dots, 0 \right),$$

$$W_{\text{sin}}^- = \text{diag} \left( \underbrace{\begin{pmatrix} 0 & 1 \\ -1 & 0 \end{pmatrix}, \dots, \begin{pmatrix} 0 & 1 \\ -1 & 0 \end{pmatrix}}_{2^{j-1} \text{ blocks}}, 0, \dots, 0 \right).$$

The above constructions of $W_{\text{sin}}^+$ and $W_{\text{sin}}^-$ satisfy,

$$W_{\text{sin}}^+ z_{\text{prev}} = \begin{bmatrix} -\sin(\omega s_1) \cos(\omega s_1) \cdots -\sin(\omega s_{2^{j-1}}) \cos(\omega s_{2^{j-1}}) \, 0 \dots 0 \end{bmatrix}^\top \in \mathbb{R}^d$$

and

$$W_{\text{sin}}^- z_{\text{prev}} = \begin{bmatrix} \sin(\omega s_1) -\cos(\omega s_1) \dots \sin(\omega s_{2^{j-1}}) -\cos(\omega s_{2^{j-1}}) \, 0 \dots 0 \end{bmatrix}^\top \in \mathbb{R}^d.$$

Each of these act blockwise on the first $2^j$ coordinates of $z_{\text{prev}}$ and zeroes out everything else in dimension $d$. We also have $z_{\text{curr}} \in \mathbb{R}^d$ with two designated coordinates $z_{\text{curr},1} = \cos(\omega d_j)$, and $z_{\text{curr},2} = \sin(\omega d_j)$, with all other coordinates being zero. We multiply $z_{\text{prev}}$ by $\cos(\omega d_j)$ and $\sin(\omega d_j)$ elementwise. Formally, the sum

$$z_{\text{curr},1} \cdot z_{\text{prev}} + z_{\text{curr},2} \cdot (M_{\text{sin}}^+ z_{\text{prev}})$$

gives the $2^{j-1}$ partial sums $\{s_k + d_j\}_{k=1}^{2^{j-1}}$ stored in the coordinates from 1 to $2^j$. We define $W_{\text{shift}} \in \mathbb{R}^{d \times d}$ in a block form with three row blocks and two column blocks:

$$W_{\text{shift}} = \begin{pmatrix} \mathbf{0}_{2^j \times 2^j} & \mathbf{0}_{2^j \times (d-2^j)} \\ \mathbf{I}_{2^j} & \mathbf{0}_{2^j \times (d-2^j)} \\ \mathbf{0}_{(d-2^{j+1}) \times 2^j} & \mathbf{0}_{(d-2^{j+1}) \times (d-2^j)} \end{pmatrix}.$$

When applied, the above matrix shifts the first $2^j$ entries of $z_{\text{prev}}$ by $2^j$ coordinates. Now, also define

$$z_{\text{curr,2}} \cdot \left( W_{\text{shift}} W_{\text{sin}}^- z_{\text{prev}} \right) + z_{\text{curr,1}} \cdot \left( W_{\text{shift}} z_{\text{prev}} \right).$$

This way, the above sum gives us the $2^{j-1}$ partial sums $\{s_k - d_j\}_{k=1}^{2^{j-1}}$ stored in the coordinates from $2^j + 1$ to $2^{j+1}$ encoded in trigonometric format. Then, we normalize this output of the model by $1/2$ and obtain the following output:

$$\begin{aligned} &\left( z_{\text{curr,1}} \cdot z_{\text{old}} + z_{\text{curr,2}} \cdot \left( M_{\text{sin}}^+ z_{\text{old}} \right) + z_{\text{curr,2}} \cdot \left( W_{\text{shift}} W_{\text{sin}}^- z_{\text{prev}} \right) + z_{\text{curr,1}} \cdot \left( W_{\text{shift}} z_{\text{prev}} \right) \right) \\ &= \left[ \cos\left( \omega \left( s_1 + d_j \right) \right), \sin\left( \omega \left( s_1 + d_j \right) \right), \ldots, \cos\left( \omega \left( s_{2^{j-1}} + d_j \right) \right), \sin\left( \omega \left( s_{2^{j-1}} + d_j \right) \right), \right. \\ &\qquad \cos\left( \omega \left( s_1 - d_j \right) \right), \sin\left( \omega \left( s_1 - d_j \right) \right), \ldots, \cos\left( \omega \left( s_{2^{j-1}} - d_j \right) \right), \sin\left( \omega \left( s_{2^{j-1}} - d_j \right) \right), \\ &\qquad 0, \ldots, 0 \right]^\top \in \mathbb{R}^d. \end{aligned}$$

Thus, this is exactly the representation of $2^j$ partial sums. This completes the argument. We should remark that, the above argument utilizes a *gated MLP* which explicitly multiplies the elements of the input features, namely, $z_{\text{curr}}$ with the partial sums $z_{\text{prev}}$. On the other hand, we don't require any nonlinear activation function, so our MLP constructions have the form $\text{MLP}(z) = W_3(W_1 z \odot W_2 z)$ for suitable choices of $W_1, W_2, W_3$ where $\odot$ denotes the Hadamard product. The use of gated MLPs is a standard practice in transformer architectures (Shazeer, 2020). □

**Proposition 7** (Read-Off MLP). *Suppose that every partial sum $s_k$ is in the range $[-S, S]$ and let $\omega < \pi/2S$. Assume that the vector*

$$z = [\cos(\omega s_1), \ \sin(\omega s_1), \ \ldots, \ \cos(\omega s_{2^n}), \ \sin(\omega s_{2^n}), 0, \ldots, 0]^\top \in \mathbb{R}^d,$$

*contains $2^n$ partial sums $\{s_1, \ldots, s_{2^n}\}$ encoded in trigonometric form, where $d = 2^{n+1} + n + 2$. Then there exists a single feed-forward network $\text{MLP}_{n+1} : \mathbb{R}^d \to \mathbb{R}^d$ such that, given input $z$, it selects the smallest nonnegative $s_\ell$ from $\{s_1, \ldots, s_{2^n}\}$ and outputs the embedding $e_{s_\ell} \in \mathbb{R}^d$, where $s_\ell$ is that minimal nonnegative partial sum.*

**Remark:** Our construction relies on gated MLP, rather than standard MLP, as in Proposition 6.

*Proof.* We know that the input embedding $z$ represents $2^n$ pairs, each pair $(\cos(\omega s_i), \sin(\omega s_i))$ stored consecutively. That is,

$$z = [\cos(\omega s_1), \ \sin(\omega s_1), \ \ldots, \ \cos(\omega s_{2^n}), \ \sin(\omega s_{2^n}), 0, \ldots, 0]^\top \in \mathbb{R}^d,$$

We will identify the smallest $s_\ell \geq 0$ and output an embedding $e_{s_\ell}$ denoting that integer. We are given that $\omega$ is small enough such that when $s_\ell \in [0, S]$, we ensure $S\omega < \pi/2$. This guarantees $\sin(\omega s_\ell) \geq 0$ if and only if $s_\ell \geq 0$. First, we wish to collapse $z$ into a single vector of size $2^n$, keeping $\cos(\omega s_\ell)$ only when $\sin(\omega s_\ell) \geq 0$ and zeroing it out otherwise. We define two matrices $W_{\text{cos}}, W_{\text{sin}} \in \mathbb{R}^{d \times d}$ by

$$\begin{aligned} (W_{\text{cos}})_{i,(2i-1)} = 1, \quad (W_{\text{cos}})_{i,j} = 0 \quad \text{for } j \neq 2i - 1, \\ (W_{\text{sin}})_{i,(2i)} = 1, \quad (W_{\text{sin}})_{i,j} = 0 \quad \text{for } j \neq 2i. \end{aligned}$$

for $1 \leq i \leq 2^n$ and all other rows/columns of $W_{\text{sin}}, W_{\text{cos}}$ are zero. Hence each matrix picks out alternate coordinates:

$$z_{\text{cos}} = W_{\text{cos}} z = \begin{bmatrix} \cos(\omega s_1) \\ \cos(\omega s_2) \\ \vdots \\ \cos(\omega s_{2^n}) \\ 0 \\ \vdots \\ 0 \end{bmatrix} \in \mathbb{R}^d, \quad z_{\text{sin}} = W_{\text{sin}} z = \begin{bmatrix} \sin(\omega s_1) \\ \sin(\omega s_2) \\ \vdots \\ \sin(\omega s_{2^n}) \\ 0 \\ \vdots \\ 0 \end{bmatrix} \in \mathbb{R}^d.$$

In order to find the minimum non-negative number, we need to find the number $s$ such that it maximizes $\cos(\omega s)$ and satisfies $\sin(\omega s) \geq 0$. For this, we utilize a sigmoid activation function in the following way:

$$z_{\text{filter}} = z_{\cos} \odot \sigma\left(c\, z_{\sin}\right),$$

where $\sigma(x) = \frac{1}{1+\exp(-x)}$ is element-wise sigmoid function, and $c \to \infty$ is a large constant. With this choice of $c$, the sigmoid output will be 1 when $s_\ell \geq 0$ and 0 otherwise. Therefore, the resulting vector $z_{\text{filter}}$ contains $\cos(\omega s)$ values at indices where $\sin(\omega s)$ is positive. Now, for $0 \leq s_\ell \leq S$ with $S\omega \leq \frac{\pi}{2}$, the ordering of $s_\ell$ from smallest to largest is the same as the ordering of $\cos(\omega s_\ell)$ from largest to smallest. Thus, to find the minimum nonnegative sum, we find the partial sum $\ell^*$ that maximizes $\cos(\omega s_\ell)$. Utilizing another gating, we calculate

$$\text{Softmax}\left(c\, z_{\text{filter}}\right)^\top z_{\text{filter}}$$

as $c \to \infty$. The softmax vector will be one-hot with 1 at index $\ell^*$ that has the largest $\cos(\omega s_\ell)$. A second multiplication with $z_{\text{filter}}$ will return this $\cos(\omega s_{\ell^*})$. Therefore, $\text{Softmax}\left(c\, z_{\text{filter}}\right)^\top z_{\text{filter}} = \cos(\omega s_{\ell^*})$. Next, we retrieve the corresponding sine entry of $s_{\ell^*}$ by applying the same one-hot selection to $z_{\sin}$. Formally,

$$\text{Softmax}\left(c\, z_{\text{filter}}\right)^\top z_{\sin} = \sin(\omega s_{\ell^*}),$$

as $c \to \infty$. Hence, from these two selected coordinates, $[\cos(\omega s_{\ell^*}), \sin(\omega s_{\ell^*})]$, we produce the final embedding in $\mathbb{R}^d$ by placing them in the first two coordinates and zeros elsewhere:

$$e_{s_{\ell^*}} = [\cos(\omega s_{\ell^*}), \sin(\omega s_{\ell^*}), 0, \ldots, 0]^\top,$$

where $s_{\ell^*}$ is the minimal nonnegative sum. This completes the argument. □

**Proposition 1** (Solving MNNS). *There exists a 1-layer transformer architecture with a mixture-of-experts MLP layer that solves the MNNS task using CoT2 by storing (sine, cosine) embeddings of all $2^k$ states at the $k$-th iteration in a non-overlapping manner.*

*Proof.* We will argue that by combining Propositions 5 to 7, we obtain a single-layer transformer that is formed by an attention block followed by an MoE feed-forward block, which solves the Minimum Non-Negative Sum (MNNS) task.

Suppose that we have $n$ input integers $d_1, \ldots, d_n$, encoded as $z_1, \ldots, z_n$, plus an arrow ($\to$) token $z_{n+1}$ and a dummy token $z_{n+2}$ corresponding to the integer 0. In this case, we will output the tokens representing the ground-truth sums $s_1, \ldots, s_n$, therefore, the number of output tokens is $m = n$ in MNNS setting. We assume that the inputs are encoded according to Assumption 2. By Proposition 6, there exist $\text{MLP}_1, \ldots, \text{MLP}_n$ that perform the following: whenever $\text{MLP}_j$ is selected with input $(z_{\text{prev}}, z_{\text{curr}})$ such that $z_{\text{prev}}$ stores $2^{j-1}$ partial sums and $z_{\text{curr}}$ stores the digit $d_j$, it adds and subtracts $d_j$ to all previously stored partial sums and stores the resulting $2^j$ partial sums in $z_{\text{out}}$. The dummy token $z_{n+2}$ that corresponds to the integer 0 allows us to initialize the partial sums from zero. If the query token is $z_{n+2}$, we produce the first partial sums by combining this dummy 0 with $d_1$, which are $(+d_1)$ and $(-d_1)$ encoded in an output token.

We assign positional encodings cyclically to output tokens. That means, the first $n+2$ input tokens have positional encodings from $p_1$ to $p_{n+2}$, and the output tokens have $p_1, p_2, \ldots$, as their positional encodings, in this exact order. This way, by Proposition 4, $\text{Attn}(z, Z)$ attends and selects the input digit tokens $z_1, z_2, \ldots, z_n$ and finally arrow $z_{n+1}$ one by one and feeds to $\text{MoEBlock}(\cdot)$.

By Proposition 5, there's a $\text{MoEBlock}(z)$ such that if the input is $z_j$ (for $j \leq n$), $\text{MLP}_j$ is selected with probability 1, and if the input is arrow token $z_{n+1}$, $\text{MLP}_{n+1}$ is selected with probability 1, which is the MLP to read-off the final answer. In the input tokens $z_1, \ldots, z_n$, the first two coordinates store the trigonometric representation of $d_1, \ldots, d_n$. To allow outputting the final answer by $\text{MLP}_{n+1}$, the partial sums obtained in the intermediate steps need to be written to separate coordinates. Therefore, $\text{MLP}_j$ takes a vector filled in the first $2^j$ coordinates, adds $d_j$ and writes to the first $2^j$ coordinates, subtracts $d_j$ and writes to the next $2^j$ coordinates, and finally divides the entire representation by 2 to maintain consistent scaling since the number of partial sums is doubled. In other words, the first $n$ MLPs have some repeated behavior. Finally, by Proposition 7, $\text{MLP}_{n+1}$ receives a vector that encodes all $2^n$ possible partial sums in cos/sin form in $2^{n+1}$ coordinates and extracts the embedding of smallest nonnegative number among them.

Altogether, this single-layer transformer with an attention module to pass the tokens to the mixture-of-experts MLP solves the Minimum Non-Negative Sum task by following CSFT described in 3. □

