# OpenReview forum: "Continuous Chain of Thought Enables Parallel Exploration and Reasoning"
_ICML.cc/2025/Workshop/TokShop — TokShop_

### Official Review · Reviewer_hEk5 · 2025-06-04
**Review of Continuous Chain of Thought paper**

**Rating:** 8
**Confidence:** 3

**Review:**

This paper introduces Continuous Chain of Thought (CoT2), a new method  with continuous  tokens for  language model's reasoning process  in place of the standard discrete token. These weighted continuous tokens allow for model to track multiple reasoning paths. The authors make several contributions like introducing supervision technique (CST), training strategy for this domain, and well established theoretical analysis that formalizes the benefits of this approach and experiments showing improved performance.
This paper is suitable for the workshop given that the paper aligns very well with fundamental concept of tokenization and well written with detailed comparison and positioning with existing work in the field.

---

### Official Review · Reviewer_TDaT · 2025-06-08
**Innovative use of multiple tokens to enable parallel reasoning paths with strong empirical gains.**

**Rating:** 8
**Confidence:** 4

**Review:**

The paper introduces a COT2 (continuous chain of thought), a new approach that extends chain of thought by using superposition of multiple tokens (i.e. continuous valued tokens) instead of discrete tokens during the reasoning process. The authors claim that this would help address the limitation of discrete sampling in language models of reduced exploration of alternative reasoning paths and increases the information capacity per token.

They also propose continuous supervised training (CSFT) to learn from these soft target (continuous valued tokens). They also explore multiple policy-optimization methods such as multi-token sampling (COT2-MTS) and Dirichlet sampling to enable self-improvement and reinforcement learning beyond the initial supervised training.

They validate their claims on one-layer transformer with COT2 to solve combinatorial problems like the subset sum problem (minimum non-negative sum, MNNS). They also quantify the benefits of COT2-MTS on inference efficiency, with improved sample complexity over discrete COT.

The authors post empirical evaluations on MNNS, ProntoQA and ProsQA, to confirm that there is significant improvement in accuracy and sample efficiency, with further gains from policy optimization.

Strengths:

1. Addressing limitation of discrete COT: The paper clearly identifies the key limitations of discrete COT, due to information bottleneck (shannon entropy, limiting the model to output at most log2(v) bit per sample, while token embeddings carry O(d) bits), and also tendency for early commitment that prematurely narrows down on reasoning paths, reducing\
 exploration. This paper addresses both these concerns by sampling a continuous superposition of tokens, thereby packing more information and tracking multiple reasoning paths parallely.
2. Strong theoretical foundation: The paper demonstrataes that single layer transformer with COT2 solved the subset sum problem significantly better, with sufficient embedding dimension. Proposition 2 shows statistical consistency of all inference method (COT2, discrete COT, and COT2-MTS) with all of them estimating the same final distribution when expectation is taken over stochastic sampling. Proposition 3 also shows that CoT2-MTR reduces total number of samples required to approximate the final distribution by a factor of K, because it can effectively track K paths in parallel.
3. New training and optimization strategies: They also introduce a new training method, Continuous supervised training (CSFT), to effectively use this new type of soft embeddings. They also go beyond and integrate GRPO based RL with multi-token sampling and dirichlet sampling to improve the performance beyond than what they achieve from the initial supervised tuning.
4. Strong Empirical results: Experiments on combinatorial type problems like MNNS, ProntoQA and ProsQA, show that the models trained using CSFT outperform discrete baselines, especially above a certain embedding dimension threshold with faster convergence. CoT2 has higher sample efficiency, where discrete COT requires multiple sampling to match one sample performance in COT2.

Weakness:
1. Interpretability: The use of superposition of multiple tokens, reduces interpretability of intermediate steps. Though already acknowledged in this paper, it would be helpful additionally, if the authors could also suggest future direction on how to probe or visualize this latent space to address this concern.
2. Applicability of Assumption 1: Assumption 1 states that next token probabilities depend only on last token and the input contex[. This assumption is well justified in MNNS, where reasoning maintains an intermediate state (the partial sum), but is this directly applicable to more complex open ended language tasks too, where a long history of explicit reasoning steps might be crucial in the next prediction too?

Justification for rating:
Why not higher: Interpretability remains a limitation, generalization to non-combinatorial task are untested, and assumption 1 needs more clarification.
Why not lower: Novel, Theoretically sound, technical depth by introducing CSFT and MST, Strong empicial gains.

---

### Decision · Program_Chairs · 2025-06-10

Accept